TOPICAL REVIEW

# Using mechanistic knowledge to appraise contemporary approaches to the rehabilitation of upper limb function following stroke

Richard G. Carson[1,2,3] ID and Kathryn S. Hayward[4,5,6] ID

[1] *Trinity College Institute of Neuroscience and School of Psychology, Trinity College Dublin, Dublin 2, Ireland*
[2] *School of Psychology, Queen's University Belfast, Belfast, UK*
[3] *School of Human Movement and Nutrition Sciences, The University of Queensland, Brisbane, Queensland, Australia*
[4] *Departments of Physiotherapy, University of Melbourne, Melbourne, Australia*
[5] *Department of Medicine, University of Melbourne, Melbourne, Australia*
[6] *The Florey, University of Melbourne, Melbourne, Australia*

Handling Editors: Laura Bennet & James Coxon

The peer review history is available in the Supporting Information section of this article (https://doi.org/10.1113/JP285559#support-information-section).

The Journal of Physiology

The Journal of **Physiology**

**Abstract** It is a paradox of neurological rehabilitation that, in an era in which preclinical models have produced significant advances in our mechanistic understanding of neural plasticity, there is inadequate support for many therapies recommended for use in clinical practice. When the goal is to estimate the probability that a specific form of therapy will have a positive clinical effect, the integration of mechanistic knowledge (concerning 'the structure or way of working

of the parts in a natural system') may improve the quality of inference. This is illustrated by analysis of three contemporary approaches to the rehabilitation of lateralized dysfunction affecting people living with stroke: constraint-induced movement therapy; mental practice; and mirror therapy. Damage to 'cross-road' regions of the structural (white matter) brain connectome generates deficits that span multiple domains (motor, language, attention and verbal/spatial memory). The structural integrity of these regions determines not only the initial functional status, but also the response to therapy. As structural disconnection constrains the recovery of functional capability, 'disconnectome' modelling provides a basis for personalized prognosis and precision rehabilitation. It is now feasible to refer a lesion delineated using a standard clinical scan to a (dis)connectivity atlas derived from the brains of other stroke survivors. As the individual disconnection pattern thus obtained suggests the functional domains most likely be compromised, a therapeutic regimen can be tailored accordingly. Stroke is a complex disorder that burdens individuals with distinct constellations of brain damage. Mechanistic knowledge is indispensable when seeking to ameliorate the behavioural impairments to which such damage gives rise.

(Received 4 October 2023; accepted after revision 12 July 2024; first published online 10 August 2024)

**Corresponding author** Richard G. Carson: Trinity College Institute of Neuroscience and School of Psychology, Trinity College Dublin, Dublin 2, Ireland.     Email: richard.carson@tcd.ie

**Abstract figure legend** In assessing potential for recovery of upper limb function following stroke, it has become customary to focus on the corticospinal tract. Damage to other regions of the structural (white matter) brain connectome generates deficits that span multiple domains (e.g. motor, language, attention and verbal/spatial memory) and determines not only initial functional status, but also the response to movement therapies. Disconnectome modelling - referring a lesion delineated using a clinical scan to a (dis)connectivity atlas derived from the brains of other stroke survivors, capitalizes upon this knowledge to provide a basis for personalized prognosis and precision rehabilitation. Abbreviations: ILF, inferior longitudinal fasciculus; SLF, superior longitudinal fasciculus. Figure redrawn and adapted from the authors' original artwork, which is available at: https://commons.wikimedia.org/wiki/File:Disconnectome_modelling.pdf. The original artwork contains elements derived from the following sources: https://plos.figshare.com/articles/figure/_3D_rendering_of_probabilistic_maps_of_the_fornix_A_the_parahippocampal_cingulum_B_the_inferior_longitudinal_fasciculus_C_and_the_superior_longitudinal_fasciculus_D_and_the_corticospinal_tract_E_and_the_uncinate_fasciculus_F_/652484 and https://www.scientificanimations.com/wp-content/uploads/2018/11/Types-of-Stroke.jpg. All artwork was published under either a CC BY 4.0 DEED license (https://creativecommons.org/licenses/by/4.0/) or Creative Commons BY-SA (Attribution-ShareAlike 4.0 International) license (https://creativecommons.org/licenses/by-sa/4.0/).

## Mechanism:

The structure or way of working of the parts in a machine or natural system.

                                        The New Shorter Oxford English Dictionary

## Introduction

There exists a profusion of approaches to the rehabilitation of upper limb function following stroke. A contemporary list might include the following: constraint-induced

**Richard Carson** is Chair in Cognitive Neuroscience of Ageing in the School of Psychology and the Institute of Neuroscience at Trinity College Dublin. He grew up near Belfast, graduated from the University of Bristol and was subsequently awarded his PhD by Simon Fraser University in 1993. He then held a series of research fellowships at the University of Queensland, before moving to Queen's University Belfast in 2006 and to Trinity College Dublin in 2011. His additional appointments are as Chair in Psychology at Queen's University Belfast and as an Honorary Professor at the University of Queensland. His research focuses upon human brain plasticity, with a particular emphasis upon changes that occur across the life-span and therapeutic approaches to the remediation of functional movement capability following brain injury. **Kate Hayward** is an Associate Professor of Stroke Recovery and Rehabilitation in the Departments of Physiotherapy and Medicine (RMH) at the University of Melbourne. She currently holds NHMRC Emerging Leadership, Heart Foundation Future Leader and Dame Kate Campbell Fellowships. Kate leads the REPAIR Research Group, which is a multidisciplinary group of clinician-scientists who seek to understand the brain–behaviour nexus during stroke recovery. Kate currently leads trials funded by NHMRC, MRFF, Heart Foundation and Stroke Foundation. She was co-Chair of the third international Stroke Recovery and Rehabilitation Roundtable.

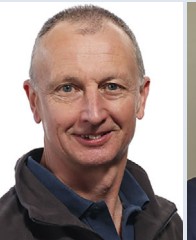
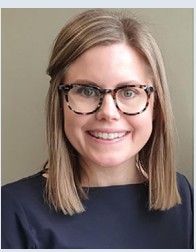

movement therapy (CIMT); bilateral arm training; neuromuscular electrical stimulation; various forms of non-invasive brain stimulation; robotic therapy; manual therapy; mental practice (MP); mirror therapy (MT); music therapy; and the use of virtual reality, among others (Pollock et al., 2014). What provides the motivation for the adoption of a specific therapy? In an era in which preclinical models have produced significant advances in our mechanistic understanding of neural plasticity (Carmichael, 2016), mechanistic knowledge is rarely the reason that interventions make their way into clinical practice. Instead, the tendency is to adopt therapies that are in vogue or that exploit new technologies, and to then seek evidence for effect, usually by focusing on the quality of the research design that links the intervention to a given outcome. The small measure of benefit derived from most new approaches to the treatment of motor dysfunction (Harvey et al., 2018; Rodgers et al., 2019) highlights the inadequacy of this approach and, perhaps, the limitations of the therapies themselves.

It is a longstanding concern (Leasure et al., 2010; Moretti et al., 2015) that information generated using preclinical models may prove insufficient to ameliorate the effects of many diseases and disorders, including stroke. There are, for example, questions that can be addressed only in our own species, as they relate to human brain networks and functions (e.g. language) and to therapies for which there are no non-human analogues. Our objective is to illustrate how mechanistic knowledge (Goodman & Gerson, 2013) derived directly from humans might be used to improve quality of inference when the goal is to estimate the probability that a specific form of therapy will have a positive clinical effect. Such knowledge can relate to mechanisms that mediate recovery from, or adaptation to, brain damage. It might also concern the means through which a rehabilitation technique exerts an effect. These forms of knowledge are integrated less frequently than might be supposed. In this presentation, we focus mainly on mechanistic physiological knowledge acquired through structural (brain) connectivity analyses. This should not be taken to suggest that other forms of mechanistic knowledge are any less important.

To be succinct, we consider three specific exemplars, namely CIMT, MP and MT. Although there is a discretionary aspect to this selection, it is used by design to illustrate more general arguments. For example, clinical practice guidelines for stroke in the UK (Intercollegiate Stroke Working Party, 2023), Australia (National Stroke Foundation Australia, 2021) and USA (Winstein et al., 2016) all promote CIMT (for 'eligible' individuals), but differ markedly with respect to their recommendation of MP and MT. Beyond illustrating that the present inferential frameworks generate variations (e.g. across developed nations) in perceived strength of evidence, it will be shown that these therapies epitomize our central point: that the utilization of mechanistic knowledge would assist in deducing the probability that each will have a positive clinical effect. Although our account is, necessarily, partial with respect to the therapies that we describe and, by extension, the individual mechanisms that are considered, it is plausible that the same general conclusions would follow if a different selection of therapies had been used instead.

## Constraint-induced therapies

**Who is likely to benefit?** In CIMT, a paretic (more impaired) limb is engaged in 'massed practice', while the other (less impaired) limb is restrained. Individuals are typically eligible for CIMT if they can actively produce at least: (i) 10° of wrist extension; (i) 10° of thumb abduction/extension; and (i) 10° of extension in at least two other digits. Approximately 10% of individuals post-stroke are likely to meet these criteria (Kwakkel et al., 2015). The efficacy of CIMT can therefore be gauged only for this small subset - a group that might benefit in equal measure from alternative therapies that offer an equivalent dose of training (Lin, Tsai et al., 2019).

In those satisfying the inclusion criteria for CIMT, the brain insult is often circumscribed in specific ways (Marumoto et al., 2013). Typically, there is preservation of the corticospinal tract (CST) projections, which, in the intact brain, traverse the posterior limb of the internal capsule and synapse onto spinal motoneurons innervating muscles located in the forearm and hand. Circuits within the primary motor cortex (M1) must also be largely preserved. Injury to M1 [and premotor (PM) cortex] is, however, trumped by insults to the CST. If damage to the CST is sufficient, the magnitude of the injury to M1–PM does not appear to have a material bearing on the recovery of upper limb function (Lin, Cloutier et al., 2019). Interestingly, CST integrity is a poor predictor of the long-term benefits of CIMT (or comparable therapy), as expressed in terms of motor ability (Wolf motor function test; Sterr et al., 2014) or Amount of Use (AOU) and Quality of Movement (QOM) (motor activity log; Sterr et al., 2014; Takebayashi et al., 2018). This might initially appear counterintuitive. For the small proportion of stroke survivors who meet the inclusion criteria for CIMT, however, there is unlikely to be significant differentiation, in terms of CST integrity, with respect to others who share this status. Thus, integrity of the CST determines category membership but does not indicate response to therapy.

**What determines the response to CIMT?** The identification of white matter tracts for which there is an association between structural integrity and level of function or course of recovery has benefited from the application of machine learning techniques to whole-brain connectomes. The initial use of these

methods, in conjunction with functional MRI, revealed characteristic abnormalities of functional (brain) connectivity following stroke (Eldaief et al., 2017). An interpretation of these disruptions as arising from damage to cortical 'hub regions' has increasingly been superseded by a recognition that in causing injury to white matter, stroke exerts its primary effects upon structural connectivity. This necessarily then gives rises to alterations in functional connectivity (Griffis et al., 2019). In particular, 'white matter bottlenecks' that disrupt functional connectivity can arise from damage to deep white matter in the frontal, temporal and parietal lobes (Griffis et al., 2020). The mapping between structural connectivity and functional outcome is sufficiently consistent to permit prognostication of the spontaneous recovery of motor capacity exhibited during the first 2 weeks post-stroke (Koch et al., 2021).

How is the significance of this mechanistic knowledge to be understood in relationship to the design and implementation of CIMT and related therapies? The key observation is that the degree to which the structural connectivity of specific white matter tracts is preserved has a bearing not only on initial functional status, but also upon the nature of the response to therapy. Along with the corpus callosum (CC; discussed below), the integrity of the superior longitudinal fasciculus (Fig. 1*A*), for example, can inform our understanding of expected short-term gains that result from training on a functional motor task (Regan et al., 2021), and responses to extended (>3 weeks) therapy (D'Imperio et al., 2021; Mattos et al., 2021). Communication within multiple brain networks is routed via the superior longitudinal fasciculus, including that which supports attention, language and praxis (Nakajima et al., 2020). Damage to this pathway imposes a significant constraint on the benefit derived from therapy. This is consistent with the supposition that effective motor recovery depends on the structural (and hence functional) integrity of distributed brain networks, extending beyond the cortex to include regions such as the insula and basal ganglia (Garcea et al., 2020).

Such dependencies should come as no surprise, given the brain networks that must be engaged to produce the repertoire of upper limb movements used in daily living, such as reaching and grasping. To undertake such actions, information concerning the characteristics of an object to be grasped, such as shape, size and orientation, derived via sensory systems, must be integrated as part of the process through which motor commands are formulated. The acquisition of motor skills is instantiated by changes in the effective connectivity of CNS circuits that concomitantly integrate somatosensory input and regulate motor output (Edwards et al., 2019). Motor learning alters both sensory and motor function (Ostry & Gribble, 2016). For individuals with stroke, the extent of sensory deficit

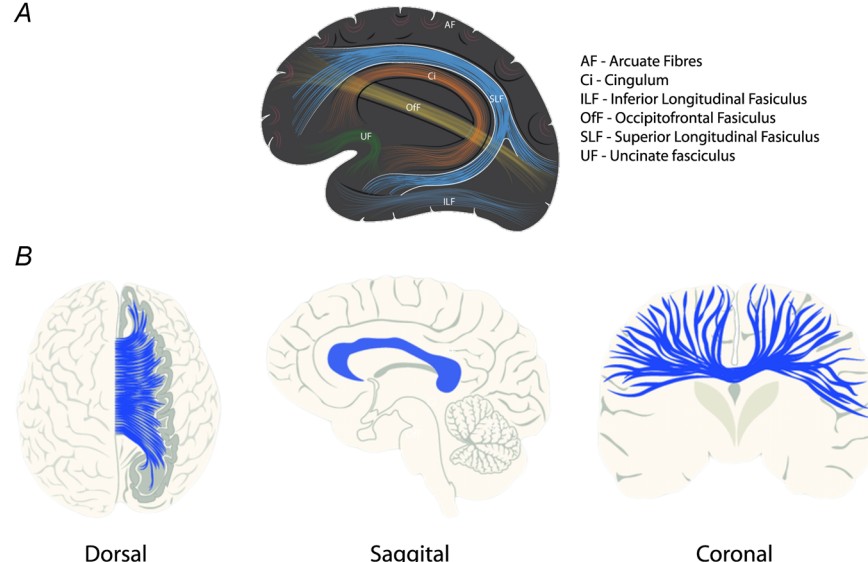

*A*

AF - Arcuate Fibres
Ci - Cingulum
ILF - Inferior Longitudinal Fasiculus
OfF - Occipitofrontal Fasiculus
SLF - Superior Longitudinal Fasiculus
UF - Uncinate fasciculus

*B*

Dorsal          Saggital          Coronal

**Figure 1. Major white matter tracts of the human brain**
*A*, fibre tracts that connect areas of the cerebral cortex located within a single hemisphere. *B*, dorsal, sagittal and coronal representations of the corpus callosum. The white matter fibres of this tract connect the two cerebral hemispheres. This artwork is available at: https://commons.wikimedia.org/wiki/File:Major_white_matter_tracts_of_the_human_brain.pdf. It contains elements derived from the following sources: https://sites.uclouvain.be/braininteratlas/en/chapter/schematic-internal-structure#slideshow-2 and https://www.researchgate.net/publication/344412345/figure/fig1/AS:961702099185703@1606299056699/The-general-organization-of-the-corpus-callosum-A-C-The-structure-and-location-of-the.png. These elements were published under a Creative Commons Attribution-Share Alike license.

constrains motor recovery (Meyer et al., 2014) and the degree of benefit that is derived from motor rehabilitation in general (Yoon et al., 2021) and CIMT in particular (Rafiei et al., 2019). The goal of an action is often to derive additional information concerning the properties of an object, once grasped. This information, in turn, supports the definition of future goals for actions. The requisite visuomotor and somatomotor transformations are mediated by parietofrontal networks and subcortical circuits in the basal ganglia and cerebellum (Errante et al., 2021). Effective specification of motor commands following brain damage depends on these distributed neural resources (Mattos et al., 2023). If the supporting structural connectome is compromised, the emergence of patterns of movement that allow for the accomplishment of task goals will also be impeded.

Such considerations are particularly relevant to the implementation of CIMT. Classically, this consists not only of restriction of the less impaired limb and intensive engagement of the more impaired limb, but also of techniques designed to promote transfer of gains derived via therapy to tasks of daily living (the 'transfer package') and a process of 'shaping' (e.g. Taub et al., 2006). The latter typically involves: (i) the frequent provision of precise feedback concerning the rapidity and quality of movement; (ii) a selection of tasks deemed appropriate to the motor deficits of the person; (iii) the use of cues, prompts and modelling as aids to performance; and (iv) increases in the level of challenge posed by the task as performance is seen to improve. We have been unable to identify any works that have sought to determine whether discernible features of brain damage impose constraints on the benefits that might arise through shaping. It is, however, readily apparent that facets (i) and (iii), at the very least, draw upon capabilities beyond those for which the integrity of efferent pathways is the determining influence. It is therefore likely that the success of the shaping element of CIMT will depend on features of the structural connectome other than those of descending projection systems, such as the CST.

**Outlook for CIMT.** Structural disconnection shapes the disruptions caused by stroke in a fundamental fashion. We suggest that it might constrain the recovery of functional capability in a similar way. What then are the implications for the design of CIMT and related therapies? As a stroke interrupts the cerebral blood flow, the brain regions perfused by the same artery are damaged in a systematic fashion. The resulting damage to white matter and subcortical regions therefore tends to cause finite clusters of correlated behavioural impairments (Corbetta et al., 2015). Thus, precision rehabilitation does not necessitate an unlimited variety of therapeutic regimens. It does, however, require sensitivity to characteristics

that define the clusters of behavioural impairment. It is, for example, apparent that the presence of multiple deficits across domains (motor, language, attention, verbal memory and spatial memory) arises from damage to 'cross-road' regions of the structural connectome that consist of several white matter tracts (in addition to specific subcortical nodes, such as the basal ganglia) (Corbetta et al., 2015). Although individuals with a level of preserved motor function sufficient to be eligible for CIMT are relatively homogeneous with respect to CST integrity, the presence of additional deficits (arising from damage to other white matter tracts) might have a significant bearing on the extent to which they are able to benefit from this (Takebayashi et al., 2018) and related forms of therapy. Reconfiguration of therapy might be necessary to ensure that deficits in non-motor domains (e.g. attention or spatial memory) do not constitute a limiting factor. Restitution of functional capacity following brain injury requires purposeful practice, for which the maintenance of motivation is a prerequisite (Lee et al., 2023). It is a persistent criticism of CIMT, as experienced by individuals with stroke, that high levels of motivation are difficult to instil (Stark et al., 2019). In this regard also, the constellation of brain damage sustained by an individual can have a significant bearing on the level of engagement with, and benefits derived from, therapy. Apathy can be modelled as a disorder arising from diffuse cerebrovascular pathology that disrupts the connectivity of large-scale networks mediating attentional control, reinforcement learning and reward-based decision-making or arising from focal damage to key nodes with these networks, such as the anterior cingulate cortex and nucleus accumbens (Tay et al., 2021). Mechanistic knowledge of the predisposing disconnection patterns might therefore be used to anticipate deficits in motivation and to design schedules of therapy that respect the capability of individual patients for sustained engagement with purposeful practice.

## Mental practice

**What is the nature of the benefit?** The term MP is applied to forms of therapy in which the patient is encouraged to rehearse mentally the execution of an action or task, without overt movement or muscle activity (Peters & Page, 2015). The rationale for its use is derived from the assumption that mental rehearsal and physical performance engage the same brain networks (Savaki & Raos, 2019). The mere fact that two processes draw (in part) upon a common subset of neural resources does not, however, imply that their sequela will be the same (Manochio et al., 2015).

In almost all clinical trials conducted with stroke survivors, MP has been included in addition to usual

therapy (Peters & Page, 2015). Thus, in meta-analyses noting a positive impact of MP, the main contrast has been usual therapy plus MP *versus* usual therapy alone (Barclay et al., 2020; Stockley et al., 2021). There is no indication that MP alone has a positive impact on upper limb function that is greater than that of usual therapy (Barclay et al., 2020). Unlike most other forms of therapy for lateralized dysfunction, the dose of MP appears to have no effect on the outcome achieved (Gaughan & Boe, 2021). This suggests that physiological pathways through which MP exerts an effect are not equivalent to those that mediate the impact of physical practice.

Mental practice fails to induce detectable changes in the activity registered in brain regions such as the supplementary motor area and cerebellum, which mediate processes of error detection and correction that are an integral part of motor learning (Kraeutner et al., 2020b). This is consistent with the hypothesis that effector-independent facets of skill acquisition (such as order of elements in a sequence) are promoted by MP to a greater degree than the development of physical capability (Ingram et al., 2016; Kraeutner et al., 2020a). There is likely to be a limit to that which can be achieved in treating upper limb dysfunction through therapies that primarily enhance effector-independent learning.

**What mediates the impact of MP?** The extent to which effector-specific effects can be induced appears to be greatest when MP precedes physical practice (Kraeutner et al., 2020a). In this vein, a brief period of 'focused concentration' on a forthcoming action both increases the level of detectable muscle activity that is present during the preparation of a grasping task and accentuates the force that is generated subsequently (Saidane et al., 2021). The more general point is that MP is associated not only with discriminable patterns of brain activity, but also with increases in postsynaptic excitability of spinal motoneurons, in both healthy adults (Bunno, 2019) and acute stroke survivors (Naseri et al., 2015). In chronic stroke survivors, alterations in reciprocal inhibition have been observed during MP (Kawakami et al., 2018). This is consistent with the anticipatory postural adjustments that are seen to accompany MP (Wider et al., 2020). Thus, MP alters the responsiveness of spinal motoneurons to synaptic input (Guggenberger et al., 2020).

Various forms of neuromuscular reorganization occur following stroke, including a loss of motor units (Marciniak et al., 2015; Li et al., 2016). These changes alter the efficiency of neural transmission, such that a given descending input produces a lower level of contractile force. In contrast, mechanisms that increase the responsiveness of spinal motoneurons can serve to amplify the behavioural response. For example, the neural adaptations induced by resistance training include a reduction in motor unit recruitment threshold and an elevation in discharge rate (Walker, 2021), both of which serve to increase the gain that relates descending neural input to mechanical muscle action. Accordingly, resistance training increases force-generating capacity and function in individuals with stroke (Veldema & Jansen, 2020).

During training conditions in which progressive increases in motor output are necessary to achieve the task goal (even as performance advances over the course of therapy), severely impaired stroke survivors exhibit elevated responsivity to corticospinal input, with reduced motor evoked potential onset latencies (correlated with improvement in arm function), even in the absence of apparent changes in descending drive from M1, and no change in motor evoked potential amplitude (Barker et al., 2012). There are ways in which the postsynaptic state of spinal motoneurons can be altered, other than through the action of direct projections from M1 (∼40% of the CST). Lesions of the CST that damage projections from M1 might leave intact those which descend from the cingulate motor area, PM cortex and supplementary motor area. Additional projection systems, including corticoreticulospinal, also have the potential to alter the state of spinal motoneurons (Calvert & Carson, 2022; Tapia et al., 2022). With particular respect to the corticoreticular tract, it has been hypothesized that this polysynaptic pathway exhibits the potential for adaptive remodelling sufficient to support some restoration of upper limb function in severely impaired stroke survivors (Choudhury et al., 2019).

A distinction is hereby drawn between remodelling and relearning. For individuals with relatively mild impairment and mostly preserved CST projections from M1, increases in functional capacity might accrue largely from relearning via purposeful practice (albeit constrained by damage to other white matter tracts and subcortical centres). Remodelling refers to the upregulation of other pathways through which brain centres alter the state of spinal motoneurons, following the loss of CST projections from M1. Given the dissimilarity of the neural adaptations that must be invoked, the characteristics of the therapeutic interventions necessary to promote relearning and remodelling are also likely to be different. That which they do have in common is the necessity for very high doses of therapy (Ballester et al., 2021). For severely impaired stroke survivors, remodelling is most likely to be promoted by extensive repetition of a relatively small number of elemental movements (Hornby et al., 2016; Levin & Demers, 2021).

**Outlook for MP.** To the extent that MP can bring about increases in the excitability of spinal motoneurons, and thus potentiate the effects of descending input, it is

conceivable that it can play a role in remodelling. We are not aware of any evidence that the effects of MP are brought about by projections other than those descending from M1 via the posterior limb of the CST. Indeed, individuals who benefit from MP are usually also able to engage in physical practice (Barclay et al., 2020). Given that, even for those with mild impairment, mental rehearsal is not superior to physical practice, it is difficult to discern a justification for its use.

## Mirror therapy

**What is the basis of the benefit?** In MT, the patient undertakes movements of the less-impaired limb while looking in a mirror that is oriented such that the reflected image gives rise to the illusion that the more-impaired limb is also moving (Thieme et al., 2018). In justifying its use in the treatment of lateralized dysfunction, most commentators have emphasized the role of visual feedback provided by the mirror. Although it can be demonstrated that the provision of such feedback gives rise to detectable changes in brain activity and increases corticospinal excitability (Reissig et al., 2014), there is little evidence that these effects induce sustained changes in functional capacity (Nogueira et al., 2021).

In all implementations of MT, the less-impaired limb performs movements repeatedly (often intensively). That this aspect of MT is given little emphasis is perhaps attributable to a concern that the deliberate use of the less-impaired limb is inconsistent with the doctrine of CIMT (Carson & Morley, 2023). In clinical trials, participants frequently receive MT in addition to usual therapy (Valkenborghs et al., 2019). Even then, when compared with control groups who are provided with an unrestricted view of the more-impaired limb, people with stroke who receive MT fail to exhibit additional improvements in motor function (Thieme et al., 2018). To our knowledge, the effect of MT has been compared directly with training of the less-impaired limb only (i.e. without the use of a mirror) in a single trial. This failed to yield an advantage for MT (Ehrensberger et al., 2019; Simpson et al., 2019). In individuals without brain injury, there has been a similar failure to obtain reliable additional effects attributable to the presence of a mirror (Chen et al., 2019).

If the mirror is not crucial (Carson & Morley, 2023), what then is the basis of the benefit that arises from engagement of the non-impaired side? It has long been recognized that motor function can be augmented via training undertaken by the opposite limb (Barss et al., 2016; Scripture et al., 1894). In addition to myriad demonstrations of this 'interlimb transfer' phenomenon in healthy individuals, similar outcomes can be obtained in people with stroke (Carson & Morley, 2023; Smyth et al.,

2023). The potential benefit of upper limb rehabilitation programmes that exploit this phenomenon is that they need not be restricted to individuals with high levels of residual function, i.e. those with preserved M1 CST projections to the more-impaired limb. Beyond the fact that those with more severe dysfunction can engage with such modes of therapy, is there a principled basis for recommending that they should do so? What are the mechanisms that mediate interlimb transfer of motor function, and are these mechanisms likely to be preserved in individuals with severe impairments?

**What mediates interlimb transfer?** Although an interceding role of additional white matter tracts cannot be precluded, the consensus view is that the CC (Fig. 1*B*) provides the anatomical substrate for the interlimb transfer of acquired proficiency (Calvert & Carson, 2022). Received wisdom is that a primary function of the CC is to inhibit activity on the opposite side of the brain (Boddington & Reynolds, 2017). Indeed, several strategies of rehabilitation using non-invasive brain stimulation are based on a mutual competition model of interactions between the cerebral hemispheres. Alternatively, it has been argued that the fundamental role of the CC is to support integrative functions that draw upon the computational resources of both cerebral hemispheres (Carson, 2020; Brancaccio et al., 2022). The corollary is that motor capability will be positively related to the structural integrity of the CC. When assessed soon (<10 days) after stroke and many months thereafter, this has been the case (Granziera et al., 2012; Hayward et al., 2017, 2022; Koh et al., 2018; Lindow et al., 2016; Mang et al., 2015).

In individuals with subcortical infarcts, changes in the structural integrity of white matter tracts in the contralesional hemisphere, encompassing the medial frontal gyrus and thalamocortical projections to primary motor, premotor and somatosensory cortices, are positively correlated with the amelioration of motor impairment during the initial 3 months postinjury (Liu et al., 2015). Similar patterns of association, extending to the structural integrity of the CC, have been noted for chronic stroke survivors who exhibit improvements in motor function in response to unilateral training of the more-impaired limb (Mattos et al., 2021). These analyses are consistent with the emerging consensus that structural disconnection, arising from damage to white matter tracts, has more significant bearing on functional connectivity within brain networks than destruction of grey matter. In this vein, the severity of the disruption to inter-hemispheric structural connectivity is associated with the level of impairment (Griffis et al., 2019) and scope for recovery (Bartolomeo, 2021) seen across multiple behavioural domains (Bartolomeo & Thiebaut de Schotten, 2016).

**Outlook for interlimb transfer.** The specific processes and structures that mediate the gains in motor function derived through interlimb transfer remain opaque (Calvert & Carson, 2022). The magnitude of transfer is, however, related to the microstructural integrity of callosal projections (Ruddy et al., 2017). Beyond a need to expand the mechanistic evidence upon which the clinical exploitation of interlimb transfer is based, there are practical questions to be addressed. On what schedule is the therapeutic potential of the interlimb transfer promoted by MT best exploited by interleaving cycles of usual therapy? Are there methods that can be applied to identify individuals who might derive the greatest benefit from this form of therapy (Kumagai et al., 2022)? In the concluding sections, we seek to address this question in a more general manner, by highlighting the potential offered by 'disconnectome' modelling as a basis for personalized prognosis and precision rehabilitation.

## Future directions: disconnectome modelling

We have shown that mechanistic knowledge provides evidence relevant to understanding the way in which a treatment might work and to determining whether a specific therapy will have a positive clinical effect. Of perhaps greater significance is the potential to use this knowledge as a basis for individually tailored therapy. Understanding of the anatomo-functional organization of the injured brain advanced first through lesion studies. Functional neuroimaging continues to provide deep insights concerning the nature of the reorganization that supports recovery. Structural connectivity analyses make plain that the location of an individual lesion is important, insofar as it engenders a specific pattern of disconnection. This determines deficits expressed across several domains (e.g. motor, sensory, cognitive and affective), the scope for functional reorganization and the likely treatment response (Guggisberg et al., 2019).

The health-care systems of even the most developed nations do not have the capacity to deploy diffusion-weighted imaging (so far used for structural connectivity analyses) for every person presenting with deficits of upper limb function following stroke. More accessible means of defining disconnection profiles to develop personalized therapy protocols might, however, be at hand. By referring the location of a patient's lesion, obtained from a conventional [e.g. $T_1/T_2$-weighted MRI, computed tomography (CT)] clinical scan, to an atlas of structural/functional connectivity derived from a database of healthy brains, the connections that would otherwise have passed through the lesion can be estimated (Smits et al., 2023). The individual disconnection patterns thus obtained are associated with deficits in various functional domains, including visual, motor, language,

attention and spatial memory (Salvalaggio et al., 2020). When the reference database is derived instead from brains of stroke survivors (Thiebaut de Schotten et al., 2020), disconnection patterns computed from individual patient lesion locations predict the motor impairments evident ≤1 year later (Bowren et al., 2022; Dulyan et al., 2022). It is possible that the prognostic power of this approach can be extended further through reference to specific task-related functional MRI meta-analytical maps (Karolis et al., 2019). The prospect offered by these advances is that, by locating a lesion (Kaffenberger et al., 2022) with respect to a suitable connectivity atlas, the functional domains most likely be disrupted can be identified and a therapeutic regimen selected accordingly. Although data-driven maps of disconnection ('disconnectomes') (Foulon et al., 2018) might not provide the level of understanding of each patient to which we ultimately aspire, they nonetheless exhibit the potential to bring about practical gains.

How might this work in practice? We consider here (Figure 2) some examples of ways in which a disconnection pattern computed from individual patient data might be used to anticipate recovery of a specific capability that is required for some forms of therapy. The reported incidence of spatial neglect following stroke is ≥30% (Hammerbeck et al., 2019). Its presence is frequently grounds for exclusion from trials of the therapies discussed herein, because it is presumed to reduce their efficacy. Yet, for almost two-thirds of such patients, their neglect symptoms are likely to resolve within 1 year (Nijboer et al., 2013). The brains of patients exhibiting severe and persisting neglect are characterized by white matter damage in regions where the second (II) and third branches of the superior longitudinal fasciculus run together and by its disruption in the splenium of the CC (Lunven et al., 2015). If these tracts are not part of a patient's disconnection profile, a phased introduction of therapies for which spatial neglect is otherwise a contra-indication can be contemplated. A prognosis that the initial deficit is likely to resolve might also prompt the use of adjuvant therapies, such as variants of non-invasive brain stimulation that appear to promote remediation of spatial neglect (Kashiwagi et al., 2018; Fan et al., 2018). With respect to apraxia, although conceptual errors in tool use will tend to dissipate, production errors are more likely to persist if the connectivity of the ventrodorsal processing stream is disrupted by damage to the supramarginal gyrus. Given that this particular disconnection pattern presages chronic deficits in a functional domain that affects tasks of daily living, specifically tailored rehabilitation strategies might be indicated (Dressing et al., 2021).

In many cases, the knowledge necessary to match efficiently a therapeutic regimen to the 'recovery connectome' (Latifi & Carmichael. 2024) does not yet exist. For example, although there are some hints

regarding structural projections (e.g. via the CC) that mediate interlimb transfer (Ruddy et al., 2017), the details of the relevant connectome are by no means clear (Calvert & Carson, 2022). Studies relating white matter integrity to poststroke cognition do, however, suggest additional ways in which the disconnection pattern of callosal fibres might inform the configuration of this or any other form of movement therapy. When assessed 3 months after stroke, measures of microstructural integrity derived for segments of the CC fibre bundle are correlated positively with global indices of cognition (Montreal Cognitive Assessment and the Mini Mental State Examination). There is a similar positive association between alterations in the microstructural integrity of callosal fibres from 3

to 12 months following stroke and changes in cognition observed over the same period (Brownsett et al., 2024). Although finer resolution of callosal projections is likely to be required, such findings point to the possibility of estimating a trajectory for post-stroke cognition sufficient to ensure that, at any given time, the demands of therapy are commensurate with the cognitive capability of the patient.

**Application of disconnectome modelling must be grounded empirically.** Analysis of disconnection patterns can be used to test assumptions that motivate specific therapies. For example, the putative role of the ventral tract in MP (Vry et al., 2012) leads to the hypothesis

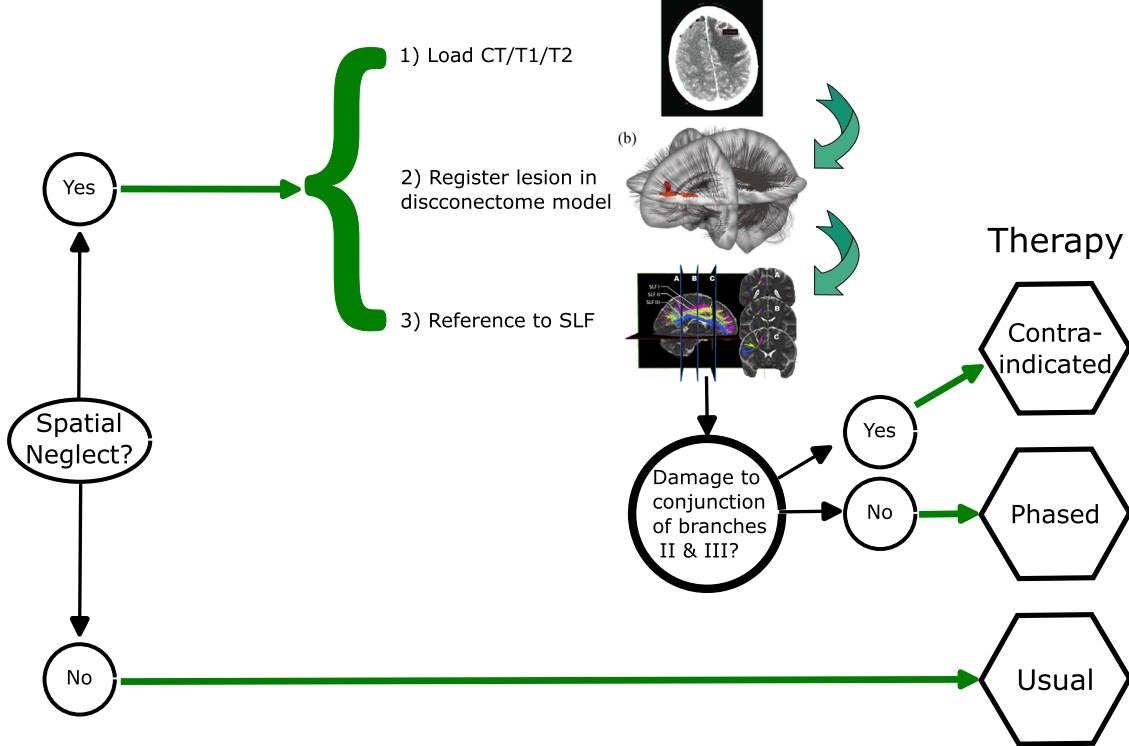

**Figure 2. Flow diagram illustrating the steps that might be taken when using disconnectome modelling as an element of the clinical decision pathway to determine the form of therapy that might be appropriate for a patient**

In the event that spatial neglect is evident, (1) provide lesion location using computed tomography (CT) or $T_1$/$T_2$-weighted MRI scan; (2) register the location of the lesion with respect to the relevant (dis)connectivity atlas; (3) determine whether the lesion location anticipates white matter damage in regions where the second (II) and third (III) branches of the superior longitudinal fasciculus (SLF) run together (or disruption in the splenium of the corpus callosum). Definitions: contra-indicated, exclude therapies for which the presence of spatial neglect is a contraindication; phased, phased introduction (over the course of ≤12 months) of therapies for which spatial neglect is otherwise a contraindication; usual, selection of the appropriate form of therapy can be based on other criteria. This artwork is available at: https://commons.wikimedia.org/wiki/File:Using_disconnectome_modelling.pdf. It contains elements derived from the following sources: https://www.researchgate.net/figure/Image-of-left-frontal-lobe-lesion-from-CT-scan-at-diagnosis-a-and-reevaluation-CT-scan_fig3_352985993 and https://www.researchgate.net/figure/Computation-of-an-individual-disconnectome-map-a-Lesion-map-A-binary-map-of-white_fig1_354980785 and https://www.frontiersin.org/files/Articles/794618/fneur-13-794618-HTML/image_m/fneur-13-794618-g002.jpg. These elements were published under a Creative Commons Attribution License (CC BY) or Creative Commons Attribution 4.0 International License.

that damage of the inferior fronto-occipital fascicule will constrain its efficacy. Without preservation of ventro-dorsal and ventral tract fibres, it has been predicted that MP will be ineffective (Hamzei et al., 2020). In theories promoting the conjecture that MP assists the acquisition of motor skill by individuals without brain injury, it has been mooted that MP co-opts neural circuits otherwise (i.e. during intended movement) engaged in goal identification and motor planning, rather than those that subserve the specification of motor 'commands' (for a review, see Hurst & Boe, 2022). Reports that frontoparietal lesions impair the facility to engage in MP (Oostra et al., 2016; McInnes et al., 2016; Sirigu et al., 1996) offer hints regarding the neural pathways required for its mediation. This apprehension is not sufficient to motivate the use of MP in rehabilitation. If it is determined that frontoparietal lesions impair MP ability, should the conclusion be drawn that MP should not be used, because individuals with brain damage of this nature lack the necessary neural resources to implement this form of therapy (McInnes et al., 2016), or should the supposition be that these individuals should be engaged in tasks that require MP, in order to promote neural adaptation sufficient to permit these cognitive operations to be supported by other brain networks? The answer to this question depends on the aspiration. If it is to enable individuals who have suffered brain injury to engage in MP, damage to frontoparietal networks might not be a contraindication. This does, however, seem an unlikely goal for rehabilitation. If, in contrast, MP is a means to an end, with the desired end being promotion of motor function, a disconnectome indicating disruption of frontoparietal networks might itself lead to the conclusion that MP will not be beneficial. Such theoretical considerations notwithstanding, and although it provides some benefit to individuals without brain injury (relative to no practice) during subsequent motor performance, MP is inferior to physical practice and has few demonstrable effects on motor learning (as indicated by retention and transfer tests) (Jiménez-Díaz et al., 2023). Indeed, for the most part, motor skills are learned without its use.

There is, however, a further, more general inference that can be made. Given that the means through which identifiable features of the connectome subserve behavioural capabilities remain largely beyond our (current) theorizing, empirically grounded (as opposed to hypothesis-driven) application of disconnectome modelling is required. Whether the modelling is restricted to classical motor networks or (as might be the case with respect to MP) encompasses other parts of the connectome, patients should not be disadvantaged, i.e. by the selection or exclusion of a particular mode of therapy, through the application of uncorroborated assumptions concerning structure–function relationships.

The ambition emphasized in most recent consensus statements is that rehabilitation be oriented towards 'motor recovery', i.e. a reduction of impairment, brought about via restitution of 'more normal patterns of motor control' (e.g. Bernhardt et al., 2017). The associated aim is to reduce the severity of symptoms such as weakness, discoordination, lack of dexterity, or ataxia. In circumstances in which restitution of normal motor control is minimal or incomplete, a degree of 'functional recovery' might yet be achieved. This can be manifested as increases in activities (of daily living) and participation (in biopsychosocial aspects of life) sufficient to conclude that the patient is doing better (O'Dell, 2023). Although the conceptual distance from connectivity analyses to treatments geared towards restitution might appear shorter than to treatments that target activities and participation, in all cases it is imperative that predictions generated through disconnectome modelling are commensurate with the aspirational goal of the rehabilitation.

**What value might be added by disconnectome modelling?** Clinical judgement is a crucial counterpoise to the application of systematized diagnostic tools (Lees, 2016). Experienced clinicians might take the view that the underlying constellation of brain damage, which will affect the choice and anticipated effect of treatment, can be inferred through careful assessment of behaviour, in other words through 'examination' of the patient. In the context of stroke care, it is, however, evident that levels of clinical competency, particularly with respect to the use of evidence-based information, are highly variable (Kipinä et al., 2024). In practice, decisions concerning access to therapy depend on subjective assessments of potential for rehabilitation (e.g. Marnane et al., 2022) and the systematic influence of factors that moderate the perception of this potential. Among these factors are professional expertise, experiential knowledge and knowledge of scientific evidence (Lam Wai Shun et al., 2017). Owing to the confluence of these effects, clinical judgements vary, and there are often disparities across even adjoining clinical centres in the prescription of rehabilitation services following stroke (e.g. Jones Berkeley et al., 2024). We contend that connectome modelling, used in conjunction with behavioural assessment, has the potential, in some specific circumstances, to democratize the basis upon which a treatment regimen is selected and administered. Access to the knowledge that might best support personalized prognosis and precision rehabilitation should not be contingent on the experience and expertise of the clinician who is assigned to a patient. In principle, personalized treatment informed by disconnection patterns can

be extended to all patients for whom a CT scan (or equivalent) is available. Of course, the degree to which approaches that incorporate connectome modelling (or, indeed, those that use only behavioural assessments) lead to improved treatment selection and outcomes must be subject to empirical determination. It is also paramount that the patient be treated, rather than the scan.

## Conclusions

Stroke is a complex disorder that burdens individuals with distinct constellations of brain damage. It demands precision rehabilitation. In this 'opinion' paper, we argue that to meet this demand, mechanistic physiological knowledge is indispensable in gauging the likelihood that a specific therapy will have a positive clinical effect, given the particular characteristics of the brain injury that has been sustained. More fundamentally, mechanistic knowledge is crucial for the formulation of therapeutic regimens that can be tailored individually to address disruptions in neural connectivity that cause deficits across multiple functional domains.

It is not being proposed that knowledge of the connectome (or disconnectome) constitutes causal–mechanistic explanation of the means through which a treatment exerts a positive effect. Indeed, such explanations can be considered qualitatively different from understanding how the material substrate described at one level supports emergent behaviour defined at another (Frégnac, 2017). Instead, our aim is to demonstrate that there is a necessary substrate for the specific physiological processes that cause material changes in behaviour brought about through rehabilitation.

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

## Additional information

### Competing interests

No competing interests declared.

## Author contributions

R.G.C. conceived the project. R.G.C. and K.S.H. wrote, revised and approved the manuscript.

## Funding

In respect of this work, the authors received no specific grant from any funding agency in the public, commercial or not-for-profit sectors. K.S.H. is supported by a National Health and Medical Research Council of Australia Emerging Leadership Fellowship (grant number: 2016420).

## Acknowledgements

## Keywords

brain imaging, corpus callosum, corticospinal, cortex, cross education, disconnectome, functional connectivity, motor, motor imagery, reticulospinal, structural connectivity, white matter

## Supporting information

Additional supporting information can be found online in the Supporting Information section at the end of the HTML view of the article. Supporting information files available:

**Peer Review History**

