## [Peer Review History · The Journal of Physiology]

Using mechanistic knowledge to appraise contemporary approaches to the rehabilitation of upper limb function following stroke.

Richard G. Carson and Kathryn S. Hayward

DOI: 10.1113/JP285559

Corresponding author(s): Richard Carson (richard.carson@tcd.ie)

Review Timeline:

Submission Date:	04-Oct-2023
Editorial Decision:	04-Dec-2023
Revision Received:	09-May-2024
Editorial Decision:	02-Jul-2024
Revision Received:	05-Jul-2024
Accepted:	12-Jul-2024

Senior Editor: Laura Bennet

Reviewing Editor: James Coxon

Transaction Report:

Dear Richard,

Re: JP-TR-2023-285559 "Using mechanistic knowledge to appraise contemporary approaches to the rehabilitation of upper limb function following stroke." by Richard G. Carson and Kathryn S. Hayward

Thank you for submitting your manuscript to The Journal of Physiology. It has been assessed by a Reviewing Editor and by 2 expert referees and we are pleased to tell you that it is potentially acceptable for publication following satisfactory major revision.

Please address all the points raised and incorporate all requested revisions or explain in your Response to Referees why a change has not been made. We hope you will find the comments helpful and that you will be able to return your revised manuscript within 3 months. If you require longer than this, please contact journal staff: jp@physoc.org. Please note that this letter does not constitute a guarantee for acceptance of your revised manuscript.

REVISION CHECKLIST:

We look forward to receiving your revised submission.

Best wishes,

Laura

Professor Laura Bennet
Senior Editor
The Journal of Physiology
<https://jp.msubmit.net>
<http://jp.physoc.org>
The Physiological Society
Hodgkin Huxley House
30 Farringdon Lane
London, EC1R 3AW
UK
<http://www.physoc.org>
<http://journals.physoc.org>

REQUIRED ITEMS

- Please include an Abstract Figure file, as well as the Figure Legend text within the main article file. The Abstract Figure is a piece of artwork designed to give readers an immediate understanding of the Review Article and should summarise the main conclusions. If possible, the image should be easily 'readable' from left to right or top to bottom. It should show the physiological relevance of the Review so readers can assess the importance and content of the article. Abstract Figures should not merely recapitulate other figures in the Review. Please try to keep the diagram as simple as possible and without superfluous information that may distract from the main conclusion of the Review. Abstract Figures must be provided by authors no later than the revised manuscript stage and should be uploaded as a separate file during online submission labelled as File Type 'Abstract Figure'. Please ensure that you include the figure legend in the main article file. All Abstract Figures will be sent to a professional illustrator for redrawing and you may be asked to approve the redrawn figure before your paper is accepted.

- Your MS must include a complete "Additional information section" with the following 4 headings and content:

Competing Interests: A statement regarding competing interests. If there are no competing interests, a statement to this effect must be included. All authors should disclose any conflict of interest in accordance with journal policy.

Author contributions: Each author should take responsibility for a particular section of the study and have contributed to writing the paper. Acquisition of funding, administrative support or the collection of data alone does not justify authorship; these contributions to the study should be listed in the Acknowledgements. Additional information such as 'X and Y have contributed equally to this work' may be added as a footnote on the title page.

It must be stated that all authors approved the final version of the manuscript and that all persons designated as authors qualify for authorship, and all those who qualify for authorship are listed.

Funding: Authors must indicate all sources of funding, including grant numbers. If authors have not received funding, this must be stated.

It is the responsibility of authors funded by RCUK to adhere to their policy regarding funding sources and underlying research material. The policy requires funding information to be included within the acknowledgement section of a paper. Guidance on how to acknowledge funding information is provided by the Research Information Network. The policy also requires all research papers, if applicable, to include a statement on how any underlying research materials, such as data, samples or models, can be accessed. However, the policy does not require that the data must be made open. If there are considered to be good or compelling reasons to protect access to the data, for example commercial confidentiality or legitimate sensitivities around data derived from potentially identifiable human participants, these should be included in the statement.

Acknowledgements: Acknowledgements should be the minimum consistent with courtesy. The wording of acknowledgements of scientific assistance or advice must have been seen and approved by the persons concerned. This section should not include details of funding.

- Author profile(s) must be uploaded via the submission form. Authors should submit a short biography (no more than 100 words for one author or 150 words in total for two authors) and a portrait photograph of the two leading authors on the paper. These should be uploaded and clearly labelled together in a Word document with the revised version of the manuscript. Any standard image format for the photograph is acceptable, but the resolution should be at least 300 DPI and preferably more. A group photograph of all authors is also acceptable, providing the biography for the whole group does not exceed 150 words.

EDITOR COMMENTS

Reviewing Editor:

This invited review has been assessed by two experts in stroke/stroke rehabilitation. Their comments are considered and insightful, and think that a considered response and revisions to the manuscript will amplify the impact of this contribution to the stroke literature.

Reviewer 2 suggests that other examples could be used. I do not think this necessary, and instead encourage further discussion of the mechanisms of the 3 chosen examples in line with the comments of both reviewers. e.g. both Reviewers identified that the mechanisms of the shaping component of CIMT should be communicated.

Reviewer 3 makes excellent points regarding the value of behavioural assessments by an experienced clinician, and that the review should give this consideration in addition to / in combination with the structural disconnection approach. This may help to position the 'value-add' of the mechanistically informed personalised treatment approach.

Reviewer 3 also points out that the disconnectome is arguably providing anatomical as opposed to physiological information. The authors might want to consider this. Are there similar atlases available where structural lesion information can be used to infer a functional disconnectome? e.g. from resting state fMRI data? Alternatively, it might just be a matter of reframing aspects of the review.

REFEREE COMMENTS

Referee #2:

Review - The Journal of Physiology

MS# JP-TR-2023-285559

'Using mechanistic knowledge to appraise...'

Thank you very much for the opportunity to provide input on this Topical Review. It is certainly an interesting review which in my opinion seeks to add volume to a growing chorus of voices proposing that we leverage our understanding of the relationship between neural insult and impairment/function to personalize approaches to rehabilitation specific to the patient. Generally, I felt the objective of the work was established in the introductory section (i.e., 'illustrate how mechanistic knowledge derived directly from humans might be used to improve quality of inference, when the goal is to estimate the probability that a specific form of therapy will have a positive clinical effect') and importantly the authors identify this mechanistic knowledge as relating to both how the brain recovers from insult and how a given therapeutic intervention works (i.e., it's purported mechanism of action).

While I have enthusiasm for the objective that the authors are seeking to achieve, I felt there were some aspects of the review that were lacking in precision and detail, and that the examples provided were not particularly illustrative of how

mechanistic knowledge could be used to personalize a therapeutic approach. On this latter point I need to offer clarity - the example provided in relation to spatial neglect was a great illustration of using knowledge of structure and function to determine a treatment pathway/approach. Rather, I was surprised that the 'How this might work in practice' section did not link back to the three therapeutic approaches discussed in the earlier sections. Moreover, the conclusion to this section of the review related to how disconnection patterns could inform on dose of therapy was vague and could use additional detail to better allow the reader to understand how we could use this information to determine what dose of therapy is required (keeping in mind that 'dose' is more than volume of repetitions, but should include discussion of frequency, intensity, time, and type of intervention).

Regarding the lack of precision, I have concern about the mechanisms underpinning CIMT and MP as they are discussed in the review. As the authors accurately point out, it is our understanding of mechanisms of both the how the brain recovers from insult and how the therapy works to achieve this. What I highlight below is a problem that we have with some therapeutic interventions that impacts applications of this disconnectome model - that is, we don't fully understand the mechanism underlying some interventions, and in some instances, the constellation of neural regions (and tracts) that would contribute to task execution is not understood or has a level of complexity that we cannot yet fully understand. I think these two points are illustrated in the MP and CIMT examples in the paper.

In this review CIMT is presented more as repetitive task practice with restraint of the unaffected limb - while this is accurate, it only captures 2/3 components of CIMT, the 3rd being the 'shaping' component. Shaping broadly referred to the behavioural component of CIMT, which involves task selection (tasks that are of interest/value to the participant to add saliency), the behavioural contract participants sign, and finally the process of the 'just right challenge', where task difficulty is regularly assessed and altered accordingly, and feedback provided to the participant immediately following each set of RTP. Indeed, Taub and colleagues identified shaping as a key component of CIMT^{1, 2}. What I am driving at here is that this element of CIMT does not appear to be accounted for in this disconnectome model as the complexity of the behaviours required goes well beyond understanding simple motor circuits (e.g., integrity of the CST). I applaud the authors for delving into the topic of motivation and how structural deficits may impact a participant's ability to engage in CIMT fully, however there are many other considerations that have not been considered. I may suggest that the authors consider more standard 'repetitive task practice' or 'use-dependent therapy' as the example here (to me they describe the same approach) as their lack of the restraint and shaping component describe their complexity and may offer a chance to better illustrate how brain lesions would impact on clinical effectiveness (and the motivation/engagement in therapy is still relevant).

As with CIMT, I felt the mechanisms underlying MP were not entirely accurate. There is a caveat to this point - I agree with the authors that generally it is understood that MP is similar to physical practice (the notion of functional equivalence and Motor Simulation Theory) which forms the basis for why we think MP helps with motor recovery. Contemporary research has revealed numerous inconsistencies in our belief that MP and PP are equivalent, and thus there are other theories that attempt to account for how MP drives motor skill acquisition. Indeed, these theories suggest MP impacts more on earlier stages of skill acquisition including goal identification and motor planning, rather than motor encoding and 'execution'.³ These different accounts of how MP work then impact on the application of the disconnectome model, in that it isn't motor pathways in isolation that would impact on clinical effectiveness, but rather other areas of the brain (for instance, numerous studies have shown fronto-parietal lesions impair MP ability 4-6 (among others)). Whether action potentials descend to the ventral horn in MP is also a point of debate, with many suggesting an inhibitory process at the level of the cortex⁷. This level of debate over mechanisms of MP is certainly too granular for such a review of course; I offer this information to highlight the issue that in many instances we don't know the mechanism of action of a therapeutic intervention, and so how we could leverage our understanding of structural/functional connectivity to personalize therapy is challenging. Similarly, the complexity of behaviours (and thus brain regions/tracts) contributing to a task are beyond our (current) understanding, and thus any approach such as this risks disadvantaging a patient based on what could be an incomplete set of data. I think this 'downside' or risk of such an approach could be mentioned. Again, and specific to the example above, the authors may want to re-consider their use of MP as an example, or alternatively could acknowledge this issue outright to provide context.

A final (and minor) comment is that I didn't find the figures to add value to the manuscript or my understanding of the concept the authors were seeking to detail. A suggestion would be to use the figures to illustrate a case study of how the disconnectome model would be applied.

Again, I appreciate the opportunity to review this paper.

REFERENCES

1. Taub E, Crago JE, Burgio LD, Groomes TE, Cook EW, 3rd, DeLuca SC et al. An operant approach to rehabilitation medicine: overcoming learned nonuse by shaping. *J Exp Anal Behav* 1994;61(2):281-93.
2. Uswatte G, Taub E, Morris D, Barman J, Crago J. Contribution of the shaping and restraint components of Constraint-Induced Movement therapy to treatment outcome. *NeuroRehabilitation* 2006;21(2):147-56.
3. Hurst AJ, Boe SG. Imagining the way forward: A review of contemporary motor imagery theory. *Frontiers in Human Neuroscience* 2022;16.
4. McInnes K, Friesen C, Boe S. Specific Brain Lesions Impair Explicit Motor Imagery Ability: A Systematic Review of the Evidence. *Arch Phys Med Rehabil* 2015.
5. Sirigu A, Duhamel J-R, Cohen L, Pillon B, et al. The mental representation of hand movements after parietal cortex damage. *Science* 1996;273(5281):1564.
6. Oostra KM, Van Bladel A, Vanhoonacker AC, Vingerhoets G. Damage to Fronto-Parietal Networks Impairs Motor Imagery Ability after Stroke: A Voxel-Based Lesion Symptom Mapping Study. *Frontiers in behavioral neuroscience* 2016;10:5.
7. Guillot A, Di Rienzo F, Macintyre T, Moran A, Collet C. Imagining is Not Doing but Involves Specific Motor Commands: A Review of Experimental Data Related to Motor Inhibition. *Front Hum Neurosci* 2012;6:247.

Referee #3:

Overall, I completely agree with the sentiment. I think there are a few areas from my perspective to give bit more thought, which could improve the manuscript a little. I do think some of the things proposed are hypotheses to be tested rather than fact, even if it seems likely to be true, so just bear that in mind.

Early on the authors state 'the utilisation of mechanistic knowledge would assist in deducing the probability that each will have a positive clinical effect.' I agree, but only if one knows what the 'active ingredient' of the treatment is, so that this can be matched up to the individual's phenotype. In other words, if a treatment is supposed to work by improving interhemispheric connectivity, you should know whether the impairments a particular individual has are due to reduced interhemispheric connectivity. The final section starts with 'We have shown that mechanistic knowledge provides evidence relevant to determining whether a specific therapy will have a positive clinical effect.' However, no mechanism of CIMT is proposed (CIMT is often termed the treatment of learned non-use). Mechanisms for MP and MT are proposed to an extent. I would say these were controversial, but perhaps this doesn't matter in the context of this article, because the authors are trying to tell us that matching mechanism (or at least which bits of brain, specifically connections, need to be intact) for the treatment to work.

As I said, in the discussion of CIMT, there is no explicit mention of its mechanism of action. It is hinted at (in reverse), when the authors say that knowing whether a patient has sensory loss, or cognitive impairment or apathy could result in altering how CIMT is delivered (although this is something experienced therapists have known all along but haven't perhaps articulated). I agree with this, but shouldn't the point be to know how a treatment is working (even if it requires multiple motor, sensory and cognitive processes), and so which anatomical pathways and/or behavioural phenotypes are required for the treatment to work, and go from there?

The reliance here seems to be on having the information on the structural brain changes first, then inferring the impaired behaviour. An experienced clinician might argue that the impaired behaviour (the thing that impacts the effect of the treatment) can be inferred from assessing the behaviour ('examining' the patient). Again, this is something therapists have done for years, without reference to the brain scan. Which approach is better is an interesting question. Neuroimaging or behaviour would need to bring an independent piece of information. The authors make a good case for this with the example of persistent neglect.

An additional point in relation to CIMT (but I accept that these are simply exemplars) is that CIMT requires 3 key elements to work - shaping, mass practice and the transfer package, each of which may have a different mechanism of action.

Re CST: 'CST integrity is a poor predictor of the long-term benefits of CIMT' but what are the long-term benefits? How are they described (impairment/activity/participation - see comments below)?

Re MP, I would suggest this is a way of 'priming' the motor system to be more responsive to subsequent motor practice. The authors discuss this in the 'What mediates impactsection'). Patients don't regain skills without moving, which is what the authors mean by 'There is likely to be a limit to that which can be achieved in treating upper limb dysfunction through therapies that primarily enhance effector independent learning.'

Re MT: I always thought MT required attempted movement of both sides, but this isn't always the protocol. The presumed mechanism of the treatment (for interlimb transfer) is suggested to depend on the corpus callosum, so that motor response will be positively related to the structural integrity of the CC. Whether one believes that interlimb transfer is the mechanism of mirror therapy, the argument made by the authors is internally consistent. However, I don't believe significant reduction of motor impairment can come about by moving the non-paretic limb.

Overall, detailed phenotyping of patients based on a combination of measured behavior and neuroimaging makes sense. The question is whether we need to know the 'mechanism of action of the treatment (to determine the likelihood of positive interaction between treatment and brain structure/connectome) for this approach to be useful.

Lastly, although I think it's probably implicit in what they are saying, all of this applies mostly to the impairment level (which is fine). Working out the individual connectome to infer likely effect of treatments targeting activity and participation seems less likely. Probably good to make this point explicitly, and from this it clearly follows how important it is to select the right type of outcome measure (if you think you are targeting impairment, or your hypothesis is that the treatment will have a differential effect on impairment depending on connectome, then you better use an impairment-based outcome measure.

p.4 'we focus mainly on mechanistic physiological knowledge acquired through structural (brain) connectivity analyses.' Isn't this anatomical knowledge?

END OF COMMENTS

Confidential Review

04-Oct-2023

The Journal of Physiology

Ms. No.: JP-TR-2023-285559

Title: Using mechanistic knowledge to appraise contemporary approaches to the rehabilitation of upper limb function following stroke.

Authors: Richard G. Carson and Kathryn S. Hayward

Response to Editors and Referees

Overview

We are grateful for the time and effort committed by the reviewers in critically evaluating our manuscript. Our responses to the concerns and questions of the Reviewing Editor and of each reviewer are presented below. In the present document, the comments received are shown in *italicized Helvetica font*. Our responses are shown in Times New Roman font. In the revised version of the manuscript, modified sections of text are shown in red font.

Reviewing Editor:

1) Reviewer 2 suggests that other examples could be used. I do not think this necessary, and instead encourage further discussion of the mechanisms of the 3 chosen examples in line with the comments of both reviewers. e.g. both Reviewers identified that the mechanisms of the shaping component of CIMT should be communicated.

In line with the recommendations of the Reviewing Editor, and the comments of both referees, we now include further consideration of the mechanisms of the three chosen examples.

With respect to Mental Practice (MP), please see our response below to Point 5 by Referee #2. In relation to the shaping component of constraint induced movement therapy (CIMT), please see our response to Point 4 by Referee #2 and to Point 5 by Referee #3. With regards to Mirror Therapy (MT), please see our response to Point 8 by Referee #3.

2) Reviewer 3 makes excellent points regarding the value of behavioural assessments by an experienced clinician, and that the review should give this consideration in addition to / in combination with the structural disconnection approach. This may help to position the 'value-add' of the mechanistically informed personalised treatment approach.

Please see our response to Point 4 by Referee #3, which reads as follows:

During the drafting phase of the original version of the paper, we discussed at length the possibility of including coverage of behavioural assessments, such as those undertaken by an experienced clinician”, with the specific aim of raising the possibility that such knowledge – if sufficiently systematised, may be used in conjunction with the information that can be derived via connectome modelling. Our conclusion was that, to do justice to this aim, the length of the manuscript would necessarily be increased substantially. Furthermore, behavioural assessments have been discussed thoroughly elsewhere.

While we therefore agree that the utilisation of behavioural assessments is critical, we hold to the view that connectome modelling has the potential to augment (rather than supplant) behavioural assessments and may, in some specific circumstances, democratize the basis upon which personalised treatment regimes can be instantiated. In principle, personalised treatment based on connectome modelling can be extended to all patients for whom a CT scan (or equivalent) is available. Of course, the degree to which approaches that incorporate connectome modelling lead to improved treatment selection and outcomes, must be subject to empirical determination.

In recognition of the importance of these points, the following section of text has been added to the Discussion section (pages 22-23 of the revised manuscript).

Clinical judgement is a crucial counterpoise to the application of systematised diagnostic tools (Lees, 2016). And experienced clinicians may take the view that the underlying constellation of brain damage – which will affect the choice and anticipated effect of treatment, may be inferred through careful assessment of behaviour, in other words, through “examination” of the patient. In the context of stroke care, it is however evident that levels of clinical competency, particularly with respect to the use of evidence-based information, are highly variable (Kipinä et al., 2024). In practice, decisions concerning access to therapy depend on subjective assessments of potential for rehabilitation (e.g., Marnane et al., 2022), and the systematic influence of factors that moderate the perception of this potential. Among these factors are professional expertise, experiential knowledge, and knowledge of scientific evidence (Lam Wai Shun et al., 2017). Due to the confluence of these effects, clinical judgements vary, and there are often disparities across even adjoining clinical centres in the prescription of rehabilitation services following stroke (e.g., Jones Berkeley et al., 2024). We contend that connectome modelling, used in conjunction with behavioural assessment, has the potential - in some specific circumstances, to democratize the basis upon which a treatment regime is selected and administered. Access to the knowledge that might best support personalised prognosis and precision rehabilitation should not be contingent on the experience and expertise of the clinician who is assigned to a patient. In principle, personalised treatment informed by disconnection patterns can be extended to all patients for whom a CT scan (or equivalent) is available. Of course, the degree to which approaches which incorporate connectome modelling (or indeed those that utilise only behavioural assessments) lead to improved treatment selection and outcomes, must be subject to empirical determination. It is also paramount that the patient be treated, rather than the scan.

3) Reviewer 3 also points out that the disconnectome is arguably providing anatomical as opposed to physiological information. The authors might want to consider this. Are there similar atlases available where structural lesion information can be used to infer a functional disconnectome? e.g. from resting state fMRI data? Alternatively, it might just be a matter of reframing aspects of the review

Please see our response to Point 11 by Referee #3, which reads as follows:

In keeping with the conception of stroke as a ‘circuitopathy’ (Cassidy et al., 2022), a decision was made to use structural (brain) connectivity analyses as means to illustrate the utilisation of mechanistic physiological knowledge. As it is the structural connectome that subserves and constrains the dynamic “functional” interactions that give rise to behaviour, it has been argued that its properties are related more fundamentally to post-stroke rehabilitation-induced

recovery (e.g., Hui et al., 2023). Aside from the challenges of inferring causal relations between functional connectivity and post-stroke recovery (Cassidy et al., 2022), the data (i.e., fMRI) that are required to derive measures of functional connectivity are not typically acquired in clinical settings.

In this context, the referee highlights that some may quibble with our assertion that structural (brain) connectivity analyses can yield “physiological knowledge”. We consider that our use of the term physiological knowledge is consistent with the scope of the term “physiology” provided by the Physiological Society, whereby, “The emphasis on integrating molecular, cellular, systems and whole body function is what distinguishes physiology from the other life sciences”. More specifically, we would contend that the functional significance of structural (brain) connectivity analyses transcends that which is understood by the term “anatomical knowledge”.

Referee #2:

1) Thank you very much for the opportunity to provide input on this Topical Review. It is certainly an interesting review which in my opinion seeks to add volume to a growing chorus of voices proposing that we leverage our understanding of the relationship between neural insult and impairment/function to personalize approaches to rehabilitation specific to the patient. Generally, I felt the objective of the work was established in the introductory section (i.e., 'illustrate how mechanistic knowledge derived directly from humans might be used to improve quality of inference, when the goal is to estimate the probability that a specific form of therapy will have a positive clinical effect') and importantly the authors identify this mechanistic knowledge as relating to both how the brain recovers from insult and how a given therapeutic intervention works (i.e., it's purported mechanism of action).

It is gratifying that the objective of the work was established clearly.

2) While I have enthusiasm for the objective that the authors are seeking to achieve, I felt there were some aspects of the review that were lacking in precision and detail, and that the examples provided were not particularly illustrative of how mechanistic knowledge could be used to personalize a therapeutic approach. On this latter point I need to offer clarity - the example provided in relation to spatial neglect was a great illustration of using knowledge of structure and function to determine a treatment pathway/approach. Rather, I was surprised that the 'How this might work in practice' section did not link back to the three therapeutic approaches discussed in the earlier sections.

In line with the recommendations of the referee, we now include further consideration of the mechanisms that may pertain to the three chosen examples.

In relation to constraint induced movement therapy (CIMT) please see our response below to Point 4. With respect to Mental Practice (MP), please see our response below to Point 5. As concerns mirror therapy (see also our response to Point 8 by Referee #3), the following additional material has been added (pages 18-19 of the revised manuscript).

In many cases, the knowledge necessary to match efficiently a therapeutic regime to the 'recovery connectome' (Latifi & Carmichael. 2024) does not yet exist. For example, although there are some hints as to structural projections (e.g., via the corpus callosum) that mediate interlimb transfer (Ruddy et al., 2017), the details of the relevant connectome are by no means clear (Calvert & Carson, 2021). Studies relating white matter integrity to poststroke cognition do however suggest additional ways in which the disconnection pattern of callosal fibres might inform the configuration of this or any other form of movement therapy. When assessed three months following stroke, measures of microstructural integrity derived for segments of the corpus callosum fibre bundle correlate positively with global indices of cognition (Montreal Cognitive Assessment and the Mini Mental State Examination). There is a similar positive association between alterations in the microstructural integrity of callosal fibres, from three to twelve months following stroke, and changes in cognition observed over the same period (Brownsett et al., 2024). Although finer resolution of callosal projections is likely to be required, such findings point to the possibility of estimating a trajectory for post stroke cognition sufficient to ensure that, at any given time, the demands of therapy are commensurate with the cognitive capability of the patient.

To make more concrete the general thrust of the approach that is being promoted – with respect to 'How this might work in practice', and as per referee's suggestion (Point 6 below), we have added an additional figure (Figure 2 in the revised manuscript).

3) Moreover, the conclusion to this section of the review related to how disconnection patterns could inform on dose of therapy was vague and could use additional detail to better allow the reader to understand how we could use this information to determine what dose of therapy is required (keeping in mind that 'dose' is more than volume of repetitions, but should include discussion of frequency, intensity, time, and type of intervention).

We accept the Referee's criticism that the section of the review relating to "how disconnection patterns could inform on dose of therapy" was weak. In this context, we acknowledge the validity of the argument that "dose' is more than volume of repetitions, but is multidimensional and should include discussion of frequency, intensity, time, and type of intervention" (e.g., Hayward et al., 2021). In view of the paucity of current evidence that supports the point we were seeking to make, and which relates specifically to the various dimensions of 'dose', we have removed the related material from the revised manuscript.

4) Regarding the lack of precision, I have concern about the mechanisms underpinning CIMT and MP as they are discussed in the review. As the authors accurately point out, it is our understanding of mechanisms of both the how the brain recovers from insult and how the therapy works to achieve this. What I highlight below is a problem that we have with some therapeutic interventions that impacts applications of this disconnectome model - that is, we don't fully understand the mechanism underlying some interventions, and in some instances, the constellation of neural regions (and tracts) that would contribute to task execution is not understood or has a level of complexity that we cannot yet fully understand. I think these two points are illustrated in the MP and CIMT examples in the paper.

In this review CIMT is presented more as repetitive task practice with restraint of the unaffected limb - while this is accurate, it only captures 2/3 components of CIMT, the 3rd being the 'shaping' component. Shaping broadly referred to the behavioural component of CIMT, which involves task selection (tasks that are of interest/value to the participant to add saliency), the behavioural contract participants sign, and finally the process of the 'just right challenge', where task difficulty is regularly assessed and altered accordingly, and feedback provided to the participant immediately following each set of RTP. Indeed, Taub and colleagues identified shaping as a key component of CIMT1, 2. What I am driving at here is that this element of CIMT does not appear to be accounted for in this disconnectome model as the complexity of the behaviours required goes well beyond understanding simple motor circuits (e.g., integrity of the CST). I applaud the authors for delving into the topic of motivation and how structural deficits may impact a participant's ability to engage in CIMT fully, however there are many other considerations that have not been considered. I may suggest that the authors consider more standard 'repetitive task practice' or 'use-dependent therapy' as the example here (to me they describe the same approach) as their lack of the restraint and shaping component describe their complexity and

may offer a chance to better illustrate how brain lesions would impact on clinical effectiveness (and the motivation/engagement in therapy is still relevant).

The Referee raises an extremely important issue, that speaks very much to the matter at hand, that is, “we don't fully understand the mechanism underlying some interventions, and in some instances, the constellation of neural regions (and tracts) that would contribute to task execution is not understood or has a level of complexity that we cannot yet fully understand”. The Referee illustrates this paucity of knowledge by invoking the 'shaping' component of CIMT (which was also mentioned by Referee #3 in their Point 5). While we appreciate that the referee may feel that we have pursued unnecessarily the more difficult option, in responding, we have opted to highlight the implications (of our lack of understanding) in the context of CIMT, rather than by using as examples “more standard 'repetitive task practice' or 'use-dependent therapy’”. Indeed, we consider that our existing point that “structural deficits may impact a participant's ability to engage in CIMT fully” can be illustrated (at least in principle) by considering the shaping component of CIMT. In this regard however, we have been unable to identify any works that have sought to determine if discernible features of brain damage impose constraints on the benefits that may arise through shaping. The issue must therefore be presented in a more abstract way.

The additional section of text (page 8 of the revised manuscript) reads as follows:

Such considerations are particularly relevant to the implementation of CIMT. Classically, this comprises not only restriction of the less impaired limb and intensive engagement of the more impaired limb, but also techniques designed to promote transfer of gains derived via therapy to tasks of daily living (the “transfer package”) and a process of “shaping” (e.g., Taub et al., 2006). The latter typically involves: 1) the frequent provision of precise feedback concerning the rapidity and quality of movement; 2) a selection of tasks deemed appropriate to the motor deficits of the person; 3) the use of cues, prompts and modelling as aids to performance; and 4) increases in the level of challenge posed by the task as performance is seen to improve. We have been unable to identify any works that have sought to determine if discernible features of brain damage impose constraints on the benefits that may arise through shaping. It is however readily apparent that facets 1 and 3, at the very least, draw upon capabilities beyond those for which the integrity of efferent pathways is the determining influence. It is likely therefore that the success of the shaping element of CIMT will depend on features of the structural connectome other than those of descending projection systems such as the CST.

5) As with CIMT, I felt the mechanisms underlying MP were not entirely accurate. There is a caveat to this point - I agree with the authors that generally it is understood that MP is similar to physical practice (the notion of functional equivalence and Motor Simulation Theory) which forms the basis for why we think MP helps with motor recovery. Contemporary research has revealed numerous inconsistencies in our belief that MP and PP are equivalent, and thus there are other theories that attempt to account for how MP drives motor skill acquisition. Indeed, these theories suggest MP impacts more on earlier stages of skill acquisition including goal identification and motor planning, rather than motor encoding and 'execution'.³ These different accounts of how MP work then impact on the application of the disconnectome model, in that it isn't motor pathways in isolation that would impact on clinical effectiveness, but rather other areas of the brain (for

instance, numerous studies have shown fronto-parietal lesions impair MP ability 4-6 (among others). Whether action potentials descend to the ventral horn in MP is also a point of debate, with many suggesting an inhibitory process at the level of the cortex⁷. This level of debate over mechanisms of MP is certainly too granular for such a review of course; I offer this information to highlight the issue that in many instances we don't know the mechanism of action of a therapeutic intervention, and so how we could leverage our understanding of structural/functional connectivity to personalize therapy is challenging. Similarly, the complexity of behaviours (and thus brain regions/tracts) contributing to a task are beyond our (current) understanding, and thus any approach such as this risks disadvantaging a patient based on what could be an incomplete set of data. I think this 'downside' or risk of such an approach could be mentioned. Again, and specific to the example above, the authors may want to re-consider their use of MP as an example, or alternatively could acknowledge this issue outright to provide context.

A principal objective of the present review was indeed to outline bases upon which knowledge could be advanced, both with respect to “relating to both how the brain recovers from insult and how a given therapeutic intervention works”. As highlighted by Referee #2: “we don't fully understand the mechanism underlying some interventions, and in some instances, the constellation of neural regions (and tracts) that would contribute to task execution is not understood or has a level of complexity that we cannot yet fully understand”. For this reason, it is not in all cases feasible to link back 'How this might work in practice' to the three therapeutic approaches discussed in the earlier sections, or at least not with high degree of confidence. Recognising that this is however an important point (and aspiration), in the Discussion section of the revised manuscript, we have sought to highlight further some of the related challenges and reinforce that the application of disconnectome modelling is not restricted to classical motor networks.

We take as the starting point for an example, the reference by Referee #2 to theories that “MP impacts more on earlier stages of skill acquisition including goal identification and motor planning, rather than motor encoding and 'execution'” (see Hurst & Boe, 2022 for a review).

The relevant section of the revised Discussion section (pages 19-20) thus reads as follows:

In theories promoting the conjecture that MP assists the acquisition of motor skill by individuals without brain injury, it has been mooted that MP co-opts neural circuits otherwise (i.e., during intended movement) engaged in goal identification and motor planning, rather than those that subserve the specification of motor “commands” (see Hurst & Boe, 2022 for a review). Reports that fronto-parietal lesions impair the facility to engage in MP (McInnes et al., 2015; Sirigu et al., 1996; Oostra et al., 2016) offer hints as to the neural pathways required for its mediation. This apprehension is not sufficient to motivate the utilisation of MP in rehabilitation. If it is determined that fronto-parietal lesions impair MP ability, should the conclusion be drawn that MP should not be employed, as individuals with brain damage of this nature lack the necessary neural resources to implement this form of therapy (McInnes et al., 2015), or, should the supposition be that these individuals should be engaged in tasks that require MP – in order to promote neural adaptation sufficient to permit these cognitive operations to be supported by other brain networks? The answer to this question depends on the aspiration. If it is to enable individuals who have suffered brain injury to engage in MP, damage to fronto-parietal networks may not be a contraindication. This does however seem an unlikely goal for rehabilitation. If on the other hand, MP is a means to an end, with the desired end being promotion of motor function, a disconnectome indicating disruption of

fronto-parietal networks, may itself lead to the conclusion that MP will not be beneficial. Such theoretical considerations notwithstanding, and though it provides some benefit to individuals without brain injury (relative to no practice) during subsequent motor performance, MP is inferior to physical practice, and has few demonstrable effects on motor learning (as indicated by retention and transfer tests) (Jiménez-Díaz et al., 2023). Indeed, for the most part, motor skills are learned without its use.

There is however a further, more general, inference that can be made. As the means through which identifiable features of the connectome subservise behavioural capabilities remain largely beyond our (current) theorising, empirically grounded (as opposed to hypothesis driven) application of disconnectome modelling is demanded. Whether the modelling is restricted to classical motor networks or (as might be the case in respect of MP) encompasses other parts of the connectome, patients should not be disadvantaged, i.e., by the selection or exclusion of a particular mode of therapy, through the application of uncorroborated assumptions concerning structure-function relations.

The referee's points that there is debate concerning "Whether action potentials descend to the ventral horn in MP" and that there are "many suggesting an inhibitory process at the level of the cortex" (Guillot et al., 2012), are well taken. Given that the analysis of MP offered in the review is with respect to its potential application in clinical settings, we believe that the summary provided by Guillot and colleagues is apposite: "with few exceptions, studies reporting a lack of EMG activity primarily investigated laboratory movements, whereas those experiments providing evidence of a muscle activity during imagery included more goal-related movements" (page 3). There are theoretical grounds upon which to question the assertion by Guillot et al. "that the motor command is actually prepared, and then blocked by inhibitory processes, during MI [motor imagery]" (see Carson, *J Physiol* 598.21 (2020) pp 4781–4802). The more prosaic consideration is that (to the best of our knowledge), when MP is employed in the context of rehabilitation, patients are never required to refrain from moving.

6) A final (and minor) comment is that I didn't find the figures to add value to the manuscript or my understanding of the concept the authors were seeking to detail. A suggestion would be to use the figures to illustrate a case study of how the disconnectome model would be applied.

We agree with the referee that Figure 1 adds little to what is already likely to be known by most readers. Indeed, many variants of this figure were generated without a satisfactory outcome being achieved. Our view is however that the original Figure 2 provides information, i.e., concerning the major white matter tracts of the human brain, that may not be accorded sufficient consideration when disruption of upper limb function is the matter at hand.

We view as excellent the referee's suggestion that a figure be used "to illustrate a case study of how the disconnectome model could be applied". This is now included as Figure 2. It takes the form of a flow diagram that includes a representation of the information that is required, the nature of the database to which reference is made, and a decision tree that conveys the manner through which a choice concerning strategies of rehabilitation might be made.

7) Again, I appreciate the opportunity to review this paper.

We take the evident commitment by the referee in critically assessing the manuscript as an indication that the goals of the paper are worthwhile.

REFERENCES

1. Taub E, Crago JE, Burgio LD, Groomes TE, Cook EW, 3rd, DeLuca SC et al. An operant approach to rehabilitation medicine: overcoming learned nonuse by shaping. *J Exp Anal Behav* 1994;61(2):281-93.
2. Uswatte G, Taub E, Morris D, Barman J, Crago J. Contribution of the shaping and restraint components of Constraint-Induced Movement therapy to treatment outcome. *NeuroRehabilitation* 2006;21(2):147-56.
3. Hurst AJ, Boe SG. Imagining the way forward: A review of contemporary motor imagery theory. *Frontiers in Human Neuroscience* 2022;16.
4. McInnes K, Friesen C, Boe S. Specific Brain Lesions Impair Explicit Motor Imagery Ability: A Systematic Review of the Evidence. *Arch Phys Med Rehabil* 2015.
5. Sirigu A, Duhamel J-R, Cohen L, Pillon B, et al. The mental representation of hand movements after parietal cortex damage. *Science* 1996;273(5281):1564.
6. Oostra KM, Van Bladel A, Vanhoonacker AC, Vingerhoets G. Damage to Fronto-Parietal Networks Impairs Motor Imagery Ability after Stroke: A Voxel-Based Lesion Symptom Mapping Study. *Frontiers in behavioral neuroscience* 2016;10:5.
7. Guillot A, Di Rienzo F, Macintyre T, Moran A, Collet C. Imagining is Not Doing but Involves Specific Motor Commands: A Review of Experimental Data Related to Motor Inhibition. *Front Hum Neurosci* 2012;6:247.

Referee #3:

1) Overall, I completely agree with the sentiment. I think there are a few areas from my perspective to give bit more thought, which could improve the manuscript a little. I do think some of the things proposed are hypotheses to be tested rather than fact, even if it seems likely to be true, so just bear that in mind.

It is gratifying the sentiments that motivated the review “struck a chord” with the referee. The constructive criticism is appreciated, and we welcome the opportunity to improve the presentation. In recognition of the specific point made here by the referee, we have amended the manuscript in several places, to emphasise that facts are not being stated. For example, we have changed (page 8 of revised manuscript):

“Structural disconnection shapes the disruptions caused by stroke in a fundamental fashion. We suggest that it constrains the recovery of functional capability in a similar way”.

to

“Structural disconnection shapes the disruptions caused by stroke in a fundamental fashion. We suggest that it may constrain the recovery of functional capability in a similar way”.

Another example is the change (pages 12-13 of the revised manuscript) of:

“Given the dissimilarity of the neural adaptations that must be invoked, the characteristics of the therapeutic interventions necessary to promote relearning and remodelling are also different”.

to

“Given the dissimilarity of the neural adaptations that must be invoked, the characteristics of the therapeutic interventions necessary to promote relearning and remodelling are also likely to be different”.

Further instances of corresponding changes are indicated by red font in the revised manuscript.

2) Early on the authors state 'the utilisation of mechanistic knowledge would assist in deducing the probability that each will have a positive clinical effect.' I agree, but only if one knows what the 'active ingredient' of the treatment is, so that this can be matched up to the individual's phenotype. In other words, if a treatment is supposed to work by improving interhemispheric connectivity, you should know whether the impairments a particular individual has are due to reduced interhemispheric connectivity. The final section starts with 'We have shown that mechanistic knowledge provides evidence relevant to determining whether a specific therapy will have a positive clinical effect.' However, no mechanism of CIMT is proposed (CIMT is often termed the treatment of learned non-use). Mechanisms for MP and MT are proposed to an extent. I would say these were controversial, but perhaps this doesn't matter in the context of this article, because the authors are trying to tell us that matching mechanism (or at least which bits of brain, specifically connections, need to be intact) for the treatment to work.

The referee's points are important and well taken. With respect to the first example provided, and the possibility that “*the impairments a particular individual has are due to reduced*

interhemispheric connectivity”, we have added new material to demonstrate possible implications (please see our response to Point 8 below and to Point 2 by Referee #2). In respect of CIMT, the challenge is indeed – as the referee highlights, that there has been no clear articulation of the mechanisms that are presumed to mediate its effects. A reading of the literature does however make apparent the widespread (although sometimes implicit) assumption that it is primarily the integrity of descending projection systems such as the CST that constrains the potency of whatever those mechanisms may be. In our critique of CIMT, we were therefore not seeking to explicate all the specific mechanisms that must play a role. Rather, our more modest aim was to emphasise that its efficacy will be contingent on the integrity of multiple brain networks, including those which support motivation, praxis and various facets of cognition including attention and language.

3) As I said, in the discussion of CIMT, there is no explicit mention of its mechanism of action. It is hinted at (in reverse), when the authors say that knowing whether a patient has sensory loss, or cognitive impairment or apathy could result in altering how CIMT is delivered (although this is something experienced therapists have known all along but haven't perhaps articulated). I agree with this, but shouldn't the point be to know how a treatment is working (even if it requires multiple motor, sensory and cognitive processes), and so which anatomical pathways and/or behavioural phenotypes are required for the treatment to work, and go from there?

We agree entirely with the point highlighted by the referee, and regret that, in the previous version of the manuscript, we were not always sufficiently explicit. To address this matter, the opening sentence of the subsection (now) labelled “Future directions: disconnectome modelling (which follows treatment of the individual therapies) now reads as follows (page 16 of the revised manuscript):

“We have shown that mechanistic knowledge provides evidence relevant **to understanding the way in which a treatment may work, and to determining** whether a specific therapy will have a positive clinical effect”.

4) The reliance here seems to be on having the information on the structural brain changes first, then inferring the impaired behaviour. An experienced clinician might argue that the impaired behaviour (the thing that impacts the effect of the treatment) can be inferred from assessing the behaviour ('examining' the patient). Again, this is something therapists have done for years, without reference to the brain scan. Which approach is better is an interesting question. Neuroimaging or behaviour would need to bring an independent piece of information. The authors make a good case for this with the example of persistent neglect.

During the drafting phase of the original version of the paper, we discussed at length the possibility of including coverage of behavioural assessments, such as those undertaken by an experienced clinician”, with the specific aim of raising the possibility that such knowledge – if sufficiently systematised, may be used in conjunction with the information that can be derived via connectome modelling. Our conclusion was that, to do justice to this aim, the length of the manuscript would necessarily be increased substantially. Furthermore, behavioural assessments have been discussed thoroughly elsewhere.

While we therefore agree that the utilisation of behavioural assessments is critical, we hold to the view that connectome modelling has the potential to augment (rather than supplant)

behavioural assessments and may, in some specific circumstances, democratize the basis upon which personalised treatment regimes can be instantiated. In principle, personalised treatment based on connectome modelling can be extended to all patients for whom a CT scan (or equivalent) is available. Of course, the degree to which approaches that incorporate connectome modelling lead to improved treatment selection and outcomes, must be subject to empirical determination.

In recognition of the importance of these points, the following section of text has been added to the Discussion section (pages 22-23 of the revised manuscript).

Clinical judgement is a crucial counterpoise to the application of systematised diagnostic tools (Lees, 2016). And experienced clinicians may take the view that the underlying constellation of brain damage – which will affect the choice and anticipated effect of treatment, may be inferred through careful assessment of behaviour, in other words, through “examination” of the patient. In the context of stroke care, it is however evident that levels of clinical competency, particularly with respect to the use of evidence-based information, are highly variable (Kipinä et al., 2024). In practice, decisions concerning access to therapy depend on subjective assessments of potential for rehabilitation (e.g., Marnane et al., 2022), and the systematic influence of factors that moderate the perception of this potential. Among these factors are professional expertise, experiential knowledge, and knowledge of scientific evidence (Lam Wai Shun et al., 2017). Due to the confluence of these effects, clinical judgements vary, and there are often disparities across even adjoining clinical centres in the prescription of rehabilitation services following stroke (e.g., Jones Berkeley et al., 2024). We contend that connectome modelling, used in conjunction with behavioural assessment, has the potential - in some specific circumstances, to democratize the basis upon which a treatment regime is selected and administered. Access to the knowledge that might best support personalised prognosis and precision rehabilitation should not be contingent on the experience and expertise of the clinician who is assigned to a patient. In principle, personalised treatment informed by disconnection patterns can be extended to all patients for whom a CT scan (or equivalent) is available. Of course, the degree to which approaches which incorporate connectome modelling (or indeed those that utilise only behavioural assessments) lead to improved treatment selection and outcomes, must be subject to empirical determination. It is also paramount that the patient be treated, rather than the scan.

5) An additional point in relation to CIMT (but I accept that these are simply exemplars) is that CIMT requires 3 key elements to work - shaping, mass practice and the transfer package, each of which may have a different mechanism of action.

In acknowledgement of this salient issue (please also see our response to Point 4 by Referee #2), the following additional section of text has been added under the heading “What determines the response to CIMT?” (page 8 of the revised manuscript):

Such considerations are particularly relevant to the implementation of CIMT. Classically, this comprises not only restriction of the less impaired limb and intensive engagement of the more impaired limb, but also techniques designed to promote transfer of gains derived via therapy to tasks of daily living (the “transfer package”) and a process of “shaping” (e.g., Taub et al., 2006). The latter typically involves: 1) the frequent provision of precise feedback concerning the rapidity and quality of movement; 2) a selection of tasks deemed appropriate to the motor deficits of the person; 3) the use of cues, prompts and modelling as aids to

performance; and 4) increases in the level of challenge posed by the task as performance is seen to improve. We have been unable to identify any works that have sought to determine if discernible features of brain damage impose constraints on the benefits that may arise through shaping. It is however readily apparent that facets 1 and 3, at the very least, draw upon capabilities beyond those for which the integrity of efferent pathways is the determining influence. It is likely therefore that the success of the shaping element of CIMT will depend on features of the structural connectome other than those of descending projection systems such as the CST.

6) *Re CST: 'CST integrity is a poor predictor of the long-term benefits of CIMT' but what are the long-term benefits? How are they described (impairment/activity/participation - see comments below)?*

We appreciate the point that greater specificity is required. The corresponding section of text (page 5 of the revised manuscript) now reads as follows:

“Interestingly, CST integrity is a poor predictor of the long-term benefits of CIMT (or comparable therapy), as expressed in terms of motor ability (Wolf Motor Function Test: Sterr et al., 2014) or Amount of Use (AOU) and Quality of Movement (QOM) (Motor Activity Log: Sterr et al., 2014; Takebayashi et al., 2018)”.

7) *Re MP, I would suggest this is a way of 'priming' the motor system to be more responsive to subsequent motor practice. The authors discuss this in the 'What mediates impactsection'. Patients don't regain skills without moving, which is what the authors mean by 'There is likely to be a limit to that which can be achieved in treating upper limb dysfunction through therapies that primarily enhance effector independent learning.'*

On this point, we believe we are in close agreement with the referee. Our note that, “The extent to which effector-specific effects can be induced appear to be greatest when MP precedes physical practice (Kraeutner *et al.*, 2020a)” is consistent with the view of the referee that MP may be “a way of 'priming' the motor system to be more responsive to subsequent motor practice”. In revising the Discussion section with this matter in mind (and in addressing Point 5 by Referee #2), we now also seek to make clear that while MP may bestow some benefit (relative to no practice) during subsequent motor performance, it is inferior to physical practice, and (at least in individuals without brain injury) has few demonstrable effects on motor learning (as indicated by retention and transfer tests) (Jiménez-Díaz et al., 2023).

The relevant section of the revised Discussion section (pages 19-20) reads as follows:

In theories promoting the conjecture that MP assists the acquisition of motor skill by individuals without brain injury, it has been mooted that MP co-opts neural circuits otherwise (i.e., during intended movement) engaged in goal identification and motor planning, rather than those that subservise the specification of motor “commands” (see Hurst & Boe, 2022 for a review). Reports that fronto-parietal lesions impair the facility to engage in MP (McInnes et al., 2015; Sirigu et al., 1996; Oostra et al., 2016) offer hints as to the neural pathways required for its mediation. This apprehension is not sufficient to motivate the utilisation of MP in rehabilitation. If it is determined that fronto-parietal lesions impair MP ability, should the conclusion be drawn that MP should not be employed, as individuals with brain damage

of this nature lack the necessary neural resources to implement this form of therapy (McInnes et al., 2015), or, should the supposition be that these individuals should be engaged in tasks that require MP – in order to promote neural adaptation sufficient to permit these cognitive operations to be supported by other brain networks? The answer to this question depends on the aspiration. If it is to enable individuals who have suffered brain injury to engage in MP, damage to fronto-parietal networks may not be a contraindication. This does however seem an unlikely goal for rehabilitation. If on the other hand, MP is a means to an end, with the desired end being promotion of motor function, a disconnectome indicating disruption of fronto-parietal networks, may itself lead to the conclusion that MP will not be beneficial. Such theoretical considerations notwithstanding, and though it provides some benefit to individuals without brain injury (relative to no practice) during subsequent motor performance, MP is inferior to physical practice, and has few demonstrable effects on motor learning (as indicated by retention and transfer tests) (Jiménez-Díaz et al., 2023). Indeed, for the most part, motor skills are acquired without its use.

There is however a further, more general, inference that can be made. As the means through which identifiable features of the connectome subservise behavioural capabilities remain largely beyond our (current) theorising, empirically grounded (as opposed to hypothesis driven) application of disconnectome modelling is demanded. Whether the modelling is restricted to classical motor networks or (as might be the case in respect of MP) encompasses other parts of the connectome, patients should not be disadvantaged, i.e., by the selection or exclusion of a particular mode of therapy, through the application of uncorroborated assumptions concerning structure-function relations.

8) Re MT: I always thought MT required attempted movement of both sides, but this isn't always the protocol. The presumed mechanism of the treatment (for interlimb transfer) is suggested to depend on the corpus callosum, so that motor response will be positively related to the structural integrity of the CC. Whether one believes that interlimb transfer is the mechanism of mirror therapy, the argument made by the authors is internally consistent. However, I don't believe significant reduction of motor impairment can come about by moving the non-paretic limb.

While the size of the extant sample is thus far small compared to other approaches to the rehabilitation of lateralized dysfunction, estimates derived through meta-analyses (Smyth *et al.*, 2023), suggest that clinically significant increases in upper limb strength and upper limb function can be brought about via interlimb transfer (i.e., through training undertaken by the less impaired limb, without the use of a mirror).

With a view to linking elements of the 'How this might work in practice' subsection below “Future directions: disconnectome modelling”, to the potential therapeutic exploitation of information concerning disconnection of callosal projections – which, it is assumed, mediate interlimb transfer, the following text has been added (pages 18-19 of the revised manuscript):

In many cases, the knowledge necessary to match efficiently a therapeutic regime to the ‘recovery connectome’ (Latifi & Carmichael, 2024) does not yet exist. For example, although there are some hints as to structural projections (e.g., via the corpus callosum) that mediate interlimb transfer (Ruddy et al., 2017), the details of the relevant connectome are by no means clear (Calvert & Carson, 2021). Studies relating white matter integrity to poststroke cognition do however suggest additional ways in which the disconnection pattern of callosal fibres might inform the configuration of this or any other form of movement therapy. When assessed three months following stroke, measures of microstructural integrity derived for

segments of the corpus callosum fibre bundle correlate positively with global indices of cognition (Montreal Cognitive Assessment and the Mini Mental State Examination). There is a similar positive association between alterations in the microstructural integrity of callosal fibres, from three to twelve months following stroke, and changes in cognition observed over the same period (Brownsett et al., 2024). Although finer resolution of callosal projections is likely to be required, such findings point to the possibility of estimating a trajectory for post stroke cognition sufficient to ensure that, at any given time, the demands of therapy are commensurate with the cognitive capability of the patient.

9) Overall, detailed phenotyping of patients based on a combination of measured behavior and neuroimaging makes sense. The question is whether we need to know the 'mechanism of action of the treatment (to determine the likelihood of positive interaction between treatment and brain structure/connectome) for this approach to be useful.

We agree fully with what we take to be the thrust of the argument being made by the referee, i.e., that which matters ultimately is the degree to which concordance between an assessment of the (dis)connectome and assignment of treatment leads ultimately to improved outcomes. In other words, it is an empirical matter. Having this in mind (please also see our response to point 7 above), the following section of text is now included as the penultimate paragraph under the subheading “Applications of disconnectome modelling must be empirically grounded” (page 20 of the revised manuscript):

“There is however a further, more general, inference that can be made. As the means through which identifiable features of the connectome subservise behavioural capabilities remain largely beyond our (current) theorising, empirically grounded (as opposed to hypothesis driven) application of disconnectome modelling is required. Whether the modelling is restricted to classical motor networks or (as might be the case in respect of MP) encompasses other parts of the connectome, patients should not be disadvantaged, i.e., by the selection or exclusion of a particular mode of therapy, through the application of uncorroborated assumptions concerning structure-function relations”.

An argument might be made that the application of disconnectome modelling can only be understood in the context of the quest for mechanistic knowledge. More saliently, to match efficiently a therapeutic regime to the ‘recovery connectome’ (Latifi & Carmichael. 2024) mechanistic knowledge concerning the “action of the treatment” is demanded. Without such knowledge, the decision space – relating the disconnectome to the treatment cannot be constrained sufficiently (to bear subsequent empirical scrutiny). Indeed, this what we take to be the thrust of the referee’s point 3.

10) Lastly, although I think it's probably implicit in what they are saying, all of this applies mostly to the impairment level (which is fine). Working out the individual connectome to infer likely effect of treatments targeting activity and participation seems less likely. Probably good to make this point explicitly, and from this it clearly follows how important it is to select the right type of outcome measure (if you think you are targeting impairment, or your hypothesis is that the treatment will have a differential effect on impairment depending on connectome, then you better use an impairment-based outcome measure.

The distinction between measures of impairment and activity is one that we have emphasised in recent works (e.g., Hayward et al., 2021), and we thank the reviewer for highlighting that it is also relevant in the context of the current submission. As we agree with the referee's suggestion that the distinction should be made explicit, and with the related point that the aspirational outcome should match the treatment mode of action, the following text has been added to the Discussion (page 21 of the revised manuscript).

The ambition emphasised in most recent consensus statements is that rehabilitation be oriented towards “motor recovery” – a reduction of impairment, brought about via restitution of “more normal patterns of motor control” (e.g., Bernhardt et al., 2017). The associated aim is to reduce the severity of symptoms such as weakness, discoordination, lack of dexterity, or ataxia. In circumstances in which restitution of normal motor control is minimal or incomplete, a degree of “functional recovery” may yet be achieved. This can be manifested as increases in activities (of daily living) and participation (in biopsychosocial aspects of life), sufficient to conclude that the patient is doing better (O'Dell, 2023). While the conceptual distance from connectivity analyses to treatments geared towards restitution may appear shorter than to treatments that target activities and participation, in all cases it is imperative that predictions generated through disconnectome modelling are commensurate with the aspirational goal of the rehabilitation.

11) p.4 'we focus mainly on mechanistic physiological knowledge acquired through structural (brain) connectivity analyses.' Isn't this anatomical knowledge?

In keeping with the conception of stroke as a “circuitopathy” (Cassidy et al., 2022), a decision was made to use structural (brain) connectivity analyses as means to illustrate the utilisation of mechanistic physiological knowledge. As it is the structural connectome that subserves and constrains the dynamic “functional” interactions that give rise to behaviour, it has been argued that its properties are related more fundamentally to post-stroke rehabilitation-induced recovery (e.g., Hui et al., 2023). Aside from the challenges of inferring causal relations between functional connectivity and post-stroke recovery (Cassidy et al., 2022), the data (i.e., fMRI) that are required to derive measures of functional connectivity are not typically acquired in clinical settings.

In this context, the referee highlights that some may quibble with our assertion that structural (brain) connectivity analyses can yield “physiological knowledge”. We consider that our use of the term physiological knowledge is consistent with the scope of the term “physiology” provided by the Physiological Society, whereby, “The emphasis on integrating molecular, cellular, systems and whole body function is what distinguishes physiology from the other life sciences”. More specifically, we would contend that the functional significance of structural (brain) connectivity analyses transcends that which is understood by the term “anatomical knowledge”.

References

- Bernhardt, J., Hayward, K. S., Kwakkel, G., Ward, N. S., Wolf, S. L., Borschmann, K., ... & Cramer, S. C. (2017). Agreed definitions and a shared vision for new standards in stroke recovery research: the stroke recovery and rehabilitation roundtable taskforce. *International Journal of Stroke*, 12(5), 444-450.
- Brownsett, S. L., Carey, L. M., Copland, D., Walsh, A., & Sihvonen, A. J. (2024). Structural brain networks correlating with poststroke cognition. *Human Brain Mapping*, 45(5), e26665.
- Cassidy, J. M., Mark, J. I., & Cramer, S. C. (2022). Functional connectivity drives stroke recovery: shifting the paradigm from correlation to causation. *Brain*, 145(4), 1211-1228.
- Hurst, A. J., & Boe, S. G. (2022). Imagining the way forward: A review of contemporary motor imagery theory. *Frontiers in human neuroscience*, 16, 1033493.
- Hui, E. S. (2023). Advanced Diffusion MRI for prediction of Stroke Recovery. *Journal of Magnetic Resonance Imaging*, 57(5), 1312-1319.
- Guillot, A., Di Rienzo, F., MacIntyre, T., Moran, A., & Collet, C. (2012). Imagining is not doing but involves specific motor commands: a review of experimental data related to motor inhibition. *Frontiers in human neuroscience*, 6, 247.
- Hayward, K. S., Kramer, S. F., Dalton, E. J., Hughes, G. R., Brodtmann, A., Churilov, L., ... & Bernhardt, J. (2021). Timing and dose of upper limb motor intervention after stroke: a systematic review. *Stroke*, 52(11), 3706-3717.
- Jiménez-Díaz, J., Chaves-Castro, K., Morera-Castro, M., Portuguez-Molina, P., & Morales-Scholz, G. (2023). Physical practice, mental practice or both: a systematic review with meta-analysis. *Journal of Physical Education and Human Movement*, 5(2), 1-14.
- Jones Berkeley, S. B., Johnson, A. M., Mormer, E. R., Ressel, K., Pastva, A. M., Wen, F., ... & Freburger, J. K. (2024). Referral to Community-Based Rehabilitation Following Acute Stroke: Findings From the COMPASS Pragmatic Trial. *Circulation: Cardiovascular Quality and Outcomes*, 17(1), e010026.
- Kipinä, P., Oikarinen, A., Mikkonen, K., Kääriäinen, M., Tuomikoski, A. M., Merilainen, M., Karsikas, E., Rantala, A., Jounila-Iloa, P., Koivunen, K. & Jarva, E. (2024). Competence of healthcare professionals in stroke care pathways: a cross-sectional study. *Journal of Vascular Nursing*.
- Lam Wai Shun, P., Bottari, C., Ogourtsova, T., & Swaine, B. (2017). Exploring factors influencing occupational therapists' perception of patients' rehabilitation potential after acquired brain injury. *Australian occupational therapy journal*, 64(2), 149-158.
- Latifi, S., & Carmichael, S. T. (2024). The emergence of multiscale connectomics-based approaches in stroke recovery. *Trends in Neurosciences*, 47(4), 303-318.
- Lees AJ. (2016). *Mentored by a madman: the William Burroughs experiment*. Devon: Notting Hill.
- McInnes, K., Friesen, C., & Boe, S. (2016). Specific brain lesions impair explicit motor imagery ability: a systematic review of the evidence. *Archives of physical medicine and rehabilitation*, 97(3), 478-489.
- O'Dell, M. W. (2023). Stroke rehabilitation and motor recovery. *CONTINUUM: Lifelong Learning in Neurology*, 29(2), 605-627.
- Oostra, K. M., Van Bladel, A., Vanhoonacker, A. C., & Vingerhoets, G. (2016). Damage to fronto-parietal networks impairs motor imagery ability after stroke: a voxel-based lesion symptom mapping study. *Frontiers in behavioral neuroscience*, 10, 5.
- Sirigu, A., Duhamel, J. R., Cohen, L., Pillon, B., Dubois, B., & Agid, Y. (1996). The mental representation of hand movements after parietal cortex damage. *Science*, 273(5281), 1564-1568.

Smyth, C., Broderick, P., Lynch, P., Clark, H., & Monaghan, K. (2023). To assess the effects of cross-education on strength and motor function in post stroke rehabilitation: a systematic literature review and meta-analysis. *Physiotherapy*, 119, 80-88.

Taub, E., Uswatte, G., King, D. K., Morris, D., Crago, J. E., & Chatterjee, A. (2006). A placebo-controlled trial of constraint-induced movement therapy for upper extremity after stroke. *Stroke*, 37(4), 1045-1049.

Dear Professor Carson,

Re: JP-TR-2024-285559R1 "Using mechanistic knowledge to appraise contemporary approaches to the rehabilitation of upper limb function following stroke." by Richard G. Carson and Kathryn S. Hayward

Thank you for submitting your manuscript to The Journal of Physiology. It has been assessed by a Reviewing Editor and by 2 expert referees and we are pleased to tell you that it is acceptable for publication following satisfactory revision.

ABSTRACT FIGURES: Authors may use The Journal's premium BioRender account to create/redraw their Abstract Figures (and any other suitable schematic figure). Information on how to access this account is here: <https://physoc.onlinelibrary.wiley.com/journal/14697793/biorender-access>.

REVISION CHECKLIST: Upload a full Response to Referees file. To create your 'Response to Referees' copy all the reports, including any comments from the Senior and Reviewing Editors, into a Microsoft Word, or similar, file and respond to each point, using font or background colour to distinguish comments and responses and upload as the required file type.

We look forward to receiving your revised submission.

Yours sincerely,

EDITOR COMMENTS

Reviewing Editor:

I have received comments on the revised manuscript from the original reviewers. Both view this work positively and appreciated the thorough approach to revising the manuscript. There is one remaining comment from Reviewer #3 for your consideration.

REFEREE COMMENTS

Referee #2:

Thanks for the opportunity to review this revised manuscript. The authors have addressed my concerns. I appreciate the care taken in addressing the points of the reviewer, and believe the manuscript is improved as a result. When a review and response unfold in this manner it reaffirms for me the impact and benefit of collegial science. Best.

Referee #3:

Thanks for thoughtful response to my comments.

I still think this is a nice opinion piece and aligns with the way I think about these things. I stress that this feels like an opinion (one I share) piece and not an authoritative review (although opinion pieces are more interesting!) A minor quibble would be that on balance it is probably too long though.

There a few other points that I return to, even though discussed by the authors. I still disagree that a disconnectome is 'mechanistic' in terms of matching treatments to patients. I think asking whether particular disconnection phenotypes respond better to different treatments is perfectly reasonable, but it doesn't tell us about the mechanism of the treatment nor the mechanism of the improvement. It's simply a 'responder' phenotype. I don't mean this as a criticism because I don't think we have to use the word mechanism in describing these relationships (difficult to show causality).

Take CIMT for example, although we could be talking about any form of repetitive task training. To benefit, there are many behaviours that need to be at least partially intact. You need a bit of movement (for which you need the CST to an extent), then you need to be able to learn (memory, sustained attention), probably better off if you don't have significant sensory loss or apraxia. I think all the authors are saying is that the structural damage measured by the disconnectome, is a short hand (and potentially quick and 'democratic') way of measuring these things in one place, since we believe in the brain structure-function relationship. I partially agree, but currently this is an idea, not something that is established or ready to role out. This article doesn't tell me what the 'CIMT-would-be-good' disconnectome is. In any case I still think it would need to be tested against the behavioural phenotype, but the authors have already responded to this point, which I made previously.

Overall, I share some of the sentiments, but I do feel they could be made more succinctly.

END OF COMMENTS

1st Confidential Review

09-May-2024

The Journal of Physiology

Ms. No.: JP-TR-2024-285559R1

Title: Using mechanistic knowledge to appraise contemporary approaches to the rehabilitation of upper limb function following stroke.

Authors: Richard G. Carson and Kathryn S. Hayward

Response to Editors and Referees

Overview

We are grateful for the time and effort committed by the reviewers in assessing the revised manuscript. Our response to the remaining comments from Referee #3 is presented below. In the present document, the comments received are shown in *italicized Helvetica font*. Our responses are shown in Times New Roman font. In the revised version of the manuscript, modified sections of text are shown in red font.

Reviewing Editor:

I have received comments on the revised manuscript from the original reviewers. Both view this work positively and appreciated the thorough approach to revising the manuscript. There is one remaining comment from Reviewer #3 for your consideration.

Referee #2:

Thanks for the opportunity to review this revised manuscript. The authors have addressed my concerns. I appreciate the care taken in addressing the points of the reviewer, and believe the manuscript is improved as a result. When a review and response unfold in this manner it reaffirms for me the impact and benefit of collegial science. Best.

We appreciate the reviewer's further positive observations.

Referee #3:

Thanks for thoughtful response to my comments.

I still think this is a nice opinion piece and aligns with the way I think about these things. I stress that this feels like an opinion (one I share) piece and not an authoritative review (although opinion pieces are more interesting!) A minor quibble would be that on balance it is probably too long though.

There a few other points that I return to, even though discussed by the authors. I still disagree that a disconnectome is 'mechanistic' in terms of matching treatments to patients. I think asking whether particular disconnection phenotypes respond better to different treatments is perfectly reasonable, but it doesn't tell us about the mechanism of the treatment nor the mechanism of the improvement. It's simply a 'responder' phenotype. I don't mean this as a criticism because I don't think we have to use the word mechanism in describing these relationships (difficult to show

causality).

Take CIMT for example, although we could be talking about any form of repetitive task training. To benefit, there are many behaviours that need to be at least partially intact. You need a bit of movement (for which you need the CST to an extent), then you need to be able to learn (memory, sustained attention), probably better off if you don't have significant sensory loss or apraxia. I think all the authors are saying is that the structural damage measured by the disconnectome, is a short hand (and potentially quick and 'democratic') way of measuring these things in one place, since we believe in the brain structure-function relationship. I partially agree, but currently this is an idea, not something that is established or ready to role out. This article doesn't tell me what the 'CIMT-would-be-good' disconnectome is. In any case I still think it would need to be tested against the behavioural phenotype, but the authors have already responded to this point, which I made previously.

Overall, I share some of the sentiments, but I do feel they could be made more succinctly.

With a view to addressing the reviewer's comments, we have taken the following steps.

1) The second sentence of the Conclusions section now commences "In this "opinion" paper, ...".

2) An additional paragraph has been added to the end of the Conclusions section. It reads as follows (page 22 of the revised manuscript):

It is not being proposed that knowledge of the connectome (or disconnectome) constitutes causal-mechanistic explanation of the means through which a treatment exerts a positive effect. Indeed, such explanations may be considered qualitatively different from understanding how the material substrate described at one level supports emergent behaviour defined at another (Frégnac, 2017). Rather, our aim is to demonstrate that there is a necessary substrate for the specific physiological processes that cause material changes in behaviour brought about through rehabilitation.

Reference

Frégnac, Y. (2017). Big data and the industrialization of neuroscience: A safe roadmap for understanding the brain? *Science*, 358(6362), 470-477.

Dear Richard,

Re: JP-TR-2024-285559R2 "Using mechanistic knowledge to appraise contemporary approaches to the rehabilitation of upper limb function following stroke." by Richard G. Carson and Kathryn S. Hayward

We are pleased to tell you that your paper has been accepted for publication in The Journal of Physiology.

Authors should note that it is too late at this point to offer corrections prior to proofing. Major corrections at proof stage, such as changes to figures, will be referred to the Editors for approval before they can be incorporated. Only minor changes, such as to style and consistency, should be made at proof stage. Changes that need to be made after proof stage will usually require a formal correction notice.

Best wishes,

Laura Bennet
Senior Editor
The Journal of Physiology

P.S. - You can help your research get the attention it deserves! Check out Wiley's free Promotion Guide for best-practice recommendations for promoting your work at www.wileyauthors.com/eeo/guide. You can learn more about Wiley Editing Services which offers professional video, design, and writing services to create shareable video abstracts, infographics, conference posters, lay summaries, and research news stories for your research at www.wileyauthors.com/eeo/promotion.

IMPORTANT NOTICE ABOUT OPEN ACCESS: To assist authors whose funding agencies mandate public access to published research findings sooner than 12 months after publication, The Journal of Physiology allows authors to pay an Open Access (OA) fee to have their papers made freely available immediately on publication.

You can check if your funder or institution has a Wiley Open Access Account here: <https://authorservices.wiley.com/author-resources/Journal-Authors/licensing-and-open-access/open-access/author-compliance-tool.html>.

EDITOR COMMENTS

Reviewing Editor:

The manuscript has been appropriately revised and there is nothing further to address.